# SPEAKERVID-5M: A LARGE-SCALE HIGH-QUALITY DATASET FOR AUDIO-VISUAL DYADIC INTERACTIVE HUMAN GENERATION

**Youliang Zhang**[1,2]  **Zhaoyang Li**[2]  **Duomin Wang**[2†]  **Jiahe Zhang**
**Deyu Zhou**[2,3]  **Zixin Yin**[2,4]  **Xili Dai**[3]  **Gang Yu**[2]  **Xiu Li**[1‡]

[1]Tsinghua University.  [2]StepFun.  [3]The Hong Kong University of Science and Technology.
zhangyou24@mails.tsinghua.edu.cn, (wangduomin,daixili.cs)@gmail.com
zyinaf@connect.ust.hk, dzhou861@connect.hkust-gz.edu.cn
yugang@stepfun.com, li.xiu@sz.tsinghua.edu.cn

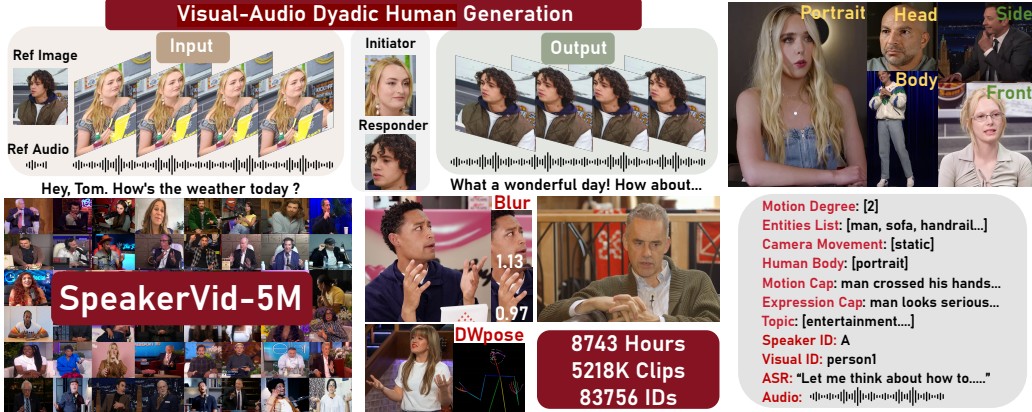

Figure 1: **Overview of the audio-visual dyadic generation task and the SpeakerVid-5M dataset.** The primary task (top row) is to generate audio-visual response based on the input of the initiator. Our SpeakerVid-5M (bottom left) provides over 8.7K hours of data to facilitate this research. Each clip is enriched with detailed multi-modal annotations (right panel), including audio, speaker ID, text labels, ASR transcript, pose sequence, hand and face clarity scores, and a full/half-body flag.

## ABSTRACT

The rapid development of large-scale models has catalyzed significant breakthroughs in the digital human domain. These advanced methodologies offer high-fidelity solutions for avatar driving and rendering, leading academia to focus on the next major challenge: audio-visual dyadic interactive virtual human. To facilitate research in this emerging area, we present **SpeakerVid-5M** dataset, the first large-scale, high-quality dataset designed for audio-visual dyadic interactive virtual human generation. Totaling over $8,743$ hours, SpeakerVid-5M contains more than $5.2$ million video clips of human portraits. It covers diverse scales and interaction types, including monadic talking, listening, and dyadic conversations. Crucially, the dataset is structured along two key dimensions: interaction type and data quality. First, it is categorized into four types (dialogue branch, single branch, listening branch and multi-turn branch) based on the interaction scenario. Second, it is stratified into a large-scale pre-training subset and a curated, high-quality subset for Supervised Fine-Tuning (SFT). This dual structure accommodates a wide array of 2D virtual human tasks. In addition, we provide an autoregressive (AR)-based video chat baseline trained on this data, accompanied by a dedicated set of metrics and test data to serve as a benchmark (**VidChatBench**) for future work. Both the dataset and the data processing code will be publicly released.

## 1 INTRODUCTION

In the era of rapid advancements in large-scale models, large video models have acquired the capability to generate high-fidelity video, presenting unprecedented advantages for the generation and driving of high-quality virtual humans. Pioneering works based on Generative Adversarial

Networks (GANs) Zhou et al. (2021); Wang et al. (2023); Yu et al. (2022), established the foundation for realistic 2D virtual human driving and rendering. However, the advent of large video models exemplified by diffusion-based works Lin et al. (2025b); Chen et al. (2025c); Wang et al. (2025b); Luo et al. (2025); Wei et al. (2025) has achieved state-of-the-art realism, significantly elevating the authenticity of both driving and rendering.

This leap in fidelity has enabled broader industrial adoption, facilitating commercial-grade applications in areas such as automated lip-syncing, digital newscasting, and virtual actors. Nevertheless, a more ambitious goal has captured the attention of researchers in both academia and industry: the creation of proactive, interactive virtual humans. This endeavor is akin to equipping an avatar with a "brain", advancing towards virtual beings that are not just passively driven but can engage autonomously. Such technology holds compelling potential for applications like more realistic virtual assistants, live-streaming e-commerce, and online education. Several works have approached this task from a system-building perspective Wang et al. (2025a); Zhu et al. (2025); Qi et al. (2025); Low & Wang (2025), have constructed functional interactive agents by integrating existing, off-the-shelf modules. In contrast to earlier systems that lack the ability for multimodal perception and understanding, BodyofHer Ao (2024) presented an end-to-end trained interactive agent based on autoregressive LLM paradigm that accepts a full suite of multimodal inputs. These works collectively represent pioneering explorations in this direction.

Training foundation models for interactive virtual humans requires vast amounts of specialized data. However, there is a notable scarcity of open-source datasets focused on interactive virtual humans within the academic community. To address this critical need, we introduce **SpeakerVid-5M**, the first large-scale, high-quality dataset for audio-visual dyadic interaction, featuring richly annotated and meticulously aligned audiovisual pairs.

SpeakerVid-5M features: (1) **Large-scale data**: The dataset contains 770K high-quality dynadic conversion audiovisual pairs (totaling 1.8K hours) and 5.2M high-quality single-speaker clips (totaling 8.7K hours). (2) **High resolution**: 93% of the videos are in 1080P or higher, ensuring detailed and clear visual input for generation tasks. (3) **Rich annotation types**: Each clip is accompanied by structured textual annotations, human skeletal sequences, automatic speech recognition (ASR) transcriptions, and blur scores (face and hand), supporting a wide range of multimodal learning objectives. (4) **High-quality**: Precise synchronization between audio and video is ensured, with rigorous filtering applied to both modalities to guarantee clean and reliable training data. (5) **Body compositional diversity**: Data instance is captured with labels spanning full-body, half-body, head-only, and side-view profiles, enabling fine-grained control over framing analysis. In addition, we select 500 videos from out-of-distribution speaker IDs to construct the **VidChatBench** benchmark for the audio-visual dyadic interactive virtual human task, which focuses on video quality, audio-visual consistency, dialogue coherence, and identity preservation for model performance evaluation.

SpeakerVid-5M is designed to facilitate research in interactive virtual human task. The rich annotations also make it a valuable resource for a variety of related tasks, such as human animation, talking head generation, multimodal dialogue modeling, etc. all of which face the challenge of a lack of large-scale, high-quality datasets. In addition, we conduct an initial exploration of implementing audio-visual dyadic interactive virtual human generation under an autoregressive paradigm. Given an input video, audio, and reference image, the model jointly generates the audio and video responses.

Our contributions can be summarized as follows:

- We propose SpeakerVid-5M, the first large-scale dataset designed specifically for the audio-visual dyadic interactive virtual human task. It includes 770K high-quality dialogue audiovisual pairs, with support for multi-turn conversations. The VidChatBench is also provided for better evaluation.

- SpeakerVid-5M contains 5M single-speaker audiovisual clips, making it the largest talking human dataset. It covers a wide range of annotated visual formats, including talking heads, half-body, full-body, and side-view videos.

- We will open-source the entire dataset, including the raw data, annotations, and data processing pipeline, providing full transparency and reproducibility for the community.

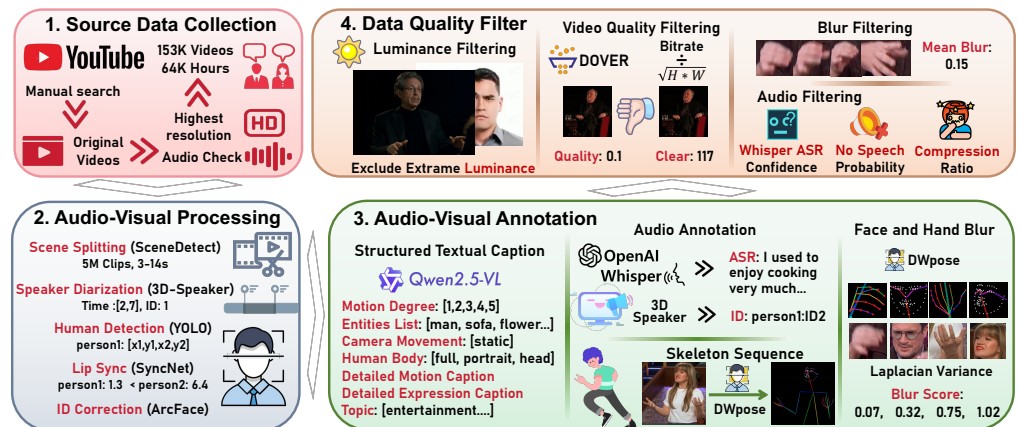

Figure 2: **The SpeakerVid-5M curation pipeline.** The process consists of: (1) Source data collection from YouTube; (2) Multi-step audio-visual pre-processing; (3) Rich multi-modal annotation using models like Qwen-VL; (4) Rigorous quality filtering stage for data fidelity.

## 2 RELATED WORK

### 2.1 AUDIO-VISUAL HUMAN VIDEO GENERATION.

Lip-sync Prajwal et al. (2020); Li et al. (2024a); Peng et al. (2025) and talking head generation Zhou et al. (2021); Wang et al. (2023); Tian et al. (2024); Yu et al. (2022); Xu et al. (2024) are foundational tasks in audio-driven human video generation, among those, PD-FGC Wang et al. (2023) was a pioneering work that first achieved the generation of talking heads with vivid facial expressions. Additionally, tasks such as learning to listen Ng et al. (2023; 2022) also fall under the category of audio-driven portrait generation, focusing on responsive non-verbal behaviors. Subsequently, with the rapid development of foundational video models, the field has bifurcated into two primary research directions. One branch has pushed for higher quality and broader scope Tian et al. (2024); Wei et al. (2025); Luo et al. (2025); Wang et al. (2025b); Chen et al. (2025c); Lin et al. (2025b), with EMO Tian et al. (2024) introducing diffusion video models to achieve state-of-the-art realism, and works like OmniHuman-1 Lin et al. (2025b), MoCha Wei et al. (2025), and Veo3 veo expanding generation to full-body, multi-agent performances, and direct text-to-audiovisual synthesis. The second branch focuses on creating interactive virtual humans Wang et al. (2025a); Zhu et al. (2025); Qi et al. (2025); Low & Wang (2025); Ao (2024). This includes modular systems like AgentAvatar Wang et al. (2025a) that integrate LLMs as planners, and end-to-end paradigms like BodyofHer Ao (2024) that enable direct multimodal understanding for autonomous reactions. These pioneering efforts in interactivity shift the focus from pure driving alignment to agent autonomy.

### 2.2 AUDIO-VISUAL HUMAN VIDEO DATASETS.

Early research in talking head and lip-sync generation initially leveraged datasets from related fields such as lip reading Afouras et al. (2018b;a); Chung & Zisserman (2017) and speaker recognition Nagrani et al. (2017); Chung et al. (2018), Subsequently, datasets specifically curated for these tasks began to emerge Yu et al. (2023); Zhang et al. (2021); Wang et al. (2021). ViCo Zhou et al. (2022) introduced a dataset for the "learning to listen" task, though it was focused on monadic scenarios. However, these early datasets were often limited in scale and of lower quality, rendering them inadequate for the demands of current high-quality, data-intensive models. While massive datasets exist, they are either too general-purpose and of variable quality (*e.g.*, ACAV-100M Lee et al. (2021)) or proprietary and unreleased (*e.g.*, the data used by OmniHuman-1 Lin et al. (2025b)). The recent release of OpenHumanVid Li et al. (2025) and TalkCuts Chen et al. (2025a) provided a valuable resource, but it was only partially released and remains focused on monadic, talking-head scenarios. This leaves a critical void for a public, large-scale dataset centered on audio-visual dyadic interactions for virtual human research. Our work, SpeakerVid-5M, is designed to fill this void.

## 3 DATASET CURATION

The construction of our SpeakerVid-5M dataset mainly includes four parts: source data collection, data pre-processing, data annotation, and quality filter. The entire process is shown in Figure 2.

## 3.1 SOURCE DATA COLLECTION

We manually collected high-quality videos featuring two-person dialogue from YouTube. The sources mainly include interviews, news reports, seminars, television programs, variety shows, debates, and educational videos. The collected videos exhibit diverse formats such as full-body, half-body, frontal, and side views, all with clear facial visibility and corresponding audio. In total, we gathered 153K audio-visual videos, amounting to $64,386$ hours of raw data. The temporal scope of the dataset extends from 2006 to June 2025. This collection spans a diverse range of genres, mainly including entertainment, people and blogs, comedy, news and politics, education, science, and sports.

## 3.2 DATA PRE-PROCESSING

**Scene splitting**. We employed SceneDetect Brandon Castellano for scene splitting. Based on the initial segmentation results, we conducted post-processing by discarding clips shorter than 3 seconds and splitting those longer than 14 seconds. As a result, we obtained video clips ranging from 3 to 14 seconds. We record the temporal order of each segment, allowing clips to be easily concatenated to form longer sequences when needed. The segments resulting from this stage are denoted by $S_{sp}$.

**Speaker diarization**. We utilize 3D-Speaker Chen et al. (2025b) to perform speaker diarization, segmenting the original audio into multiple segments and recording the timestamps and speaker IDs for each segment. For each original audio, we identified the two primary speaker IDs based on the frequency and duration of their speech. The segments resulting from this stage are denoted by $S_{sv}$.

**Human detection**. We apply YOLO Jocher & the Ultralytics Team (2023) for human tracking within each video clip to form a spatiotemporal visual track for each individual. We then refine each clip by temporally and spatially cropping based on YOLO results to obtain single-speaker video clips. This stage may yield multiple clips, each originating from different spatial regions within the same temporal window of the single video clip $S_{sp}$. The resulting segments are denoted by $S_{rsp}$.

**Lip synchronization**. For each video clip $S_{rsp}$, we first calculate the temporal overlap between $S_{rsp}$ and $S_{sv}$, which yields the overlapped segment $S_{ol}$. We then employ SyncNet Raina & Arora (2022) to perform audio-visual synchronization within each segment $S_{ol}$. The SyncNet confidence score is used to associate each speaker ID with a specific visual bounding box. In cases where two individuals are present, we assign speaker ID to the bounding box with the highest confidence score.

**ID correction.** To further verify and refine the speaker IDs obtained through speaker diarization and lip-sync, we employ ArcFace Deng et al. (2019a) for further correction. For multiple clips extracted from the same original video, the speaker IDs derived from audio analysis should be consistent with visual analysis results. For speaker IDs from audio analysis, we compute facial cosine similarity across clips with the same speaker ID. Outliers with low similarity scores are compared with other IDs using ArcFace. Outliers will be corrected if a higher similarity is found with another ID.

## 3.3 DATA ANNOTATION

**Structured Textual Caption.** To comprehensively capture the video content, we generate structurally detailed textual annotations using the advanced multimodal model Qwen2.5-VL Bai et al. (2025), which is renowned for its proficiency in visual understanding. These annotations encompass camera movement, a list of present entities, body orientation (front or side), subject framing (half- or full-body), and detailed descriptions of body movements and facial expressions. Concurrently, we employ Qwen-3 Yang et al. (2025) to summarize the ASR transcripts from multiple clips of the same source video, which informs our annotation of the dialogue's topic category.

**Audio Annotation.** Each data sample in our dataset consists of a speaker video with well-aligned audio and visual modalities. To support tasks related to audio generation and control, we annotate the audio with several key features. First, we apply Whisper Radford et al. (2023) for automatic speech recognition to obtain text transcriptions of the audio. Second, we annotate each audio segment with its corresponding SyncNet metrics. Finally, we use 3D-Speaker to assign speaker IDs to multiple audio segments from a single original audio. In addition to the original audio, we provide a cleaned version in which segments not belonging to the target speaker ID are replaced with silent audio.

**Skeleton Sequence.** Building upon the spatiotemporal human bounding boxes obtained from YOLO, we further utilize DWpose Yang et al. (2023) for human pose estimation. The resulting skeletal sequences include face, hands, and body. Clips without a detected face are filtered out.

**Blur Score.** We derive face and hand bounding boxes from DWpose annotations. For each frame, these regions are cropped and resized to a $128 \times 128$ resolution. We then compute the Laplacian variance of each crop, which serves as an image clarity score, with higher values indicating greater sharpness. Given that rapid limb movements often induce motion blur, incorporating this clarity score as a conditioning factor can enhance the model's performance in such scenarios Lin et al. (2025a).

**Motion score.** The degree of human motion is a crucial, yet highly subjective and variable, element in human-centered videos. To address this, we employed Qwen2.5-VL to score the magnitude of motion on a 1-to-5 scale (1=minimal, 5=high amplitude). To simulate the diverse perspectives of different annotators, multiple distinct persona-based prompts were used for each video. The final motion annotation was calculated by averaging the scores after excluding statistical outliers.

### 3.4 QUALITY FILTER

**Luminance Filtering.** Following OpenHumanVid Li et al. (2025), we calculate the luminance score to filter overly dark or bright videos. The luminance score is calculated using the formula: $0.2126R + 0.7152G + 0.0722B$, where $R$, $G$, and $B$ denote the pixel values of the red, green, and blue channels, respectively. Videos with a luminance score below 10 or above 210 are filtered out.

**Video Quality Filtering.** To ensure the visual quality of the video data, we employed the DOVER Wu et al. (2023) model for video quality assessment. DOVER decomposes each video into aesthetics-related and technical-related components and evaluates them separately. This approach enhances the consistency between the model's assessment results and human subjective perception of video quality. We filtering out videos with fused scores below 0.25.

**Clear score Filtering.** Resolution alone is insufficient to assess the visual clarity of a video, as some high-resolution videos may still exhibit blurriness and other low-quality characteristics. Bitrate reflects the amount of information carried per frame and provides an informative quality measure. We compute a clarity score as $B/\sqrt{W \times H}$, where $W$ and $H$ means the resolution of a video and $B$ is the bitrate. Videos with clarity scores in the bottom $5\%$ are filtered out.

**Blur Filtering.** Human-centered videos often suffer from motion blur, especially when the subject exhibits large movements. In our dataset, such blur typically affects the face and hands. To address this, we compute the average blur score over face and hand for each frame in a video. Videos with an average blur score(face or hand) below 0.1 are filtered out to ensure data quality.

**Audio Filtering.** Previous audio-visual datasets have typically focused solely on visual quality while neglecting the quality of the audio. We additionally focus on four key metrics in the ASR process: confidence score, no-speech probability, compression ratio, and detected language. These indicators reflect the reliability of the ASR output, whether the audio contains human speech, and whether there are corrupted or non-speech signals in the audio. We filter out audio clips with a confidence score(average log probability) lower than $-1.5$, a no-speech probability greater than 0.8, or a compression ratio exceeding 2.5. To further ensure the quality of the detected results, we filter out samples in which the detected language mismatches the language labeled in the video's metadata.

## 4 DATASET COMPARISON AND ANALYSIS

### 4.1 DATASET STATISTICAL COMPARISON.

As shown in Table 1, we present a comparative analysis of our dataset against previous general and human video datasets. Compared to traditional audio- or text-driven digital human generation, our dataset is the first to extend this task into audio-visual dyadic interactive scenarios, featuring complete audio-visual pairs of both questions and responses. This provides rich, high-quality training data for end-to-end audio-visual dyadic interactive virtual human generation. Moreover, our dataset represents the largest collection of single-speaker audio-video pairs to date. Each clip features well-aligned audio and video, capturing a clear and uninterrupted instance of speech.

| Datasets | Domain | Clips | Duration (hours) | Generation | Person num | audio | pose | Speaker ID | Blur anno | Body composition | Caption type | IDs | Resolution |
|---|---|---|---|---|---|---|---|---|---|---|---|---|---|
| UCF-101 Soomro et al. (2012) | Human | 13.3K | 26.7 | – | N/A | | | | | | Text | N/A | 240P |
| ActivityNet Caba Heilbron et al. (2015) | Human | 100K | 849 | – | N/A | | | | | | Text | N/A | N/A |
| NTU RGB+D Shahroudy et al. (2016) | Human | 114K | 3.7 | Conditioned | single | | ✓ | | | | - | N/A | 1080P |
| TikTok-v4 Chang et al. (2023) | Human | 350 | 1 | Conditioned | single | | ✓ | | | | - | N/A | N/A |
| Openhumanvid Li et al. (2025) | Human | 13.4M | 16.7K | Conditioned | multi | ✓ | ✓ | | | | Structured | N/A | 720P |
| VoxCeleb Nagrani et al. (2017) | Head | 21.2K | 352 | Conditioned | single | ✓ | | | | | - | 1.2k | 224P |
| VoxCeleb2 Chung et al. (2018) | Head | 150.4K | 2.4K | Conditioned | single | ✓ | | | | | - | 6.1K | 224P |
| MEAD Wang et al. (2020) | Head | 281.4K | 39 | Conditioned | single | ✓ | | | | | - | 60 | 1080P |
| CelebV-HQ Zhu et al. (2022) | Head | 35.6K | 68 | Conditioned | single | ✓ | | | | | Structured | 15.6K | 512P |
| CelebV-Text Yu et al. (2023) | Head | 70K | 279 | Conditioned | single | ✓ | | | | | Structured | N/A | 512P |
| SpeakerVid-5M | Human | 5.2M | 8.7K | Conditioned | single | ✓ | ✓ | ✓ | ✓ | ✓ | Structured | 83K | 1080P |
| SpeakerVid-5M (Dialogue) | Human | 770K | 1.8K | Dyadic | single | ✓ | ✓ | ✓ | ✓ | ✓ | Structured | 16K | 1080P |

Table 1: **Comparative analysis of SpeakerVid-5M with existing human video datasets.** Person num indicates the number of individuals in a given clip. Blur anno represents the degree of blurriness of the hands and face in each frame. Body composition means the fine-grained annotations for body composition (full-body, half-body, head-only) and camera perspective (frontal, side), features that are absent in most prior work.

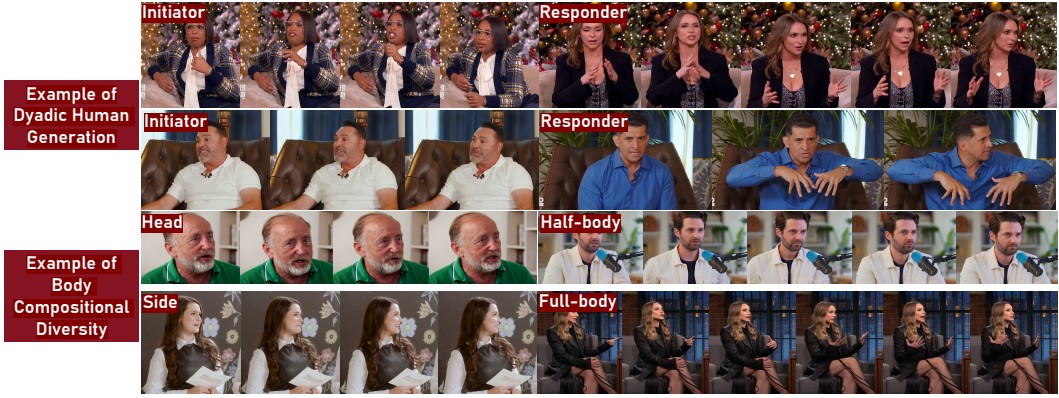

Figure 3: **Examples of dyadic dialogue and body composition in SpeakerVid-5M.** The top rows illustrate a typical dyadic human generation sample (initiator and responder). The bottom rows demonstrate the variety of body compositions annotated in our dataset, including close-up headshots, half-body, and full-body views, which are critical for controllable generation.

Key features of our dataset include: **(1) Large-scale data**. The dataset comprises 5.2M audio-video clips totaling 8.7K hours, and 770K two-person conversational audiovisual pairs totaling 1.8K hours. **(2) High resolution**. 93% of the videos are in 1080P or higher, and 98% exceed 720P, ensuring high visual fidelity. **(3) Rich annotations**. Each clip is accompanied by fine-grained structured textual annotations, ASR transcriptions, human pose sequences, motion magnitude scores, and motion blur scores, enabling detailed modeling and analysis. **(4) Diverse data formats**. The dataset includes single-speaker monologues, two-person dialogues, multi-turn conversational dialogues, and listening-human scenarios, supporting a wide range of human-centric tasks. **(5) Tiered dataset design:** The dataset is organized into a large-scale pretraining subset and a curated high-quality subset (1.3K hours), facilitating research under varying levels of training resources and computational constraints. **(6) Body compositional diversity**: Data instance is captured with labels spanning full-body, half-body, head, and side-view profiles, enabling fine-grained control over framing analysis.

## 4.2 SPEAKERVID-5M DIALOGUE BRANCH

To support the training of audio-visual dyadic interactive virtual human generation, we introduce a two-person dialogue branch within the dataset, where each sample consists of two audio-visual pairs, one serving as the input and the other as the target response. Unlike condition-controlled generation tasks such as talking head, or monadic generation tasks such as learning-to-listen, dyadic generation requires the model to generate both audio and video responses based on a comprehensive understanding of the input multi-modal content. This setting goes beyond conventional modality-aligned generation, demanding stronger comprehension and reasoning capabilities from generative models. This branch consists of 770K clip pairs (1.8K hours) and includes 16K unique speaker IDs. The dialogue topics span a wide range of categories, including entertainment, people and blogs, comedy, news and politics, education, and science.

## 4.3 SPEAKERVID-5M SINGLE BRANCH

The single-speaker video branch consists of 5.2M clips with 83K unique speaker IDs, totaling 8.7K hours. It covers a diverse range of camera framings and angles, including full-body, half-body, frontal,

and side-view shots. With respect to its scale, this branch constitutes the largest talking human dataset to date, with a volume on par with major general-purpose video datasets.

## 4.4 SPEAKERVID-5M LISTENING BRANCH

For practical applications, digital human models are required not only to generate meaningful responses but also to exhibit appropriate listening behaviors. Therefore, we specifically collect two types of listening data: **(1) Co-present listening dialogue** (*e.g.*, live interviews, news commentary, seminars). For dialogues featuring two individuals on-screen, we identify listening segments using audio segmentation and SyncNet confidence scores. Given a clip with two individuals $A$ and $B$ (before cropping), if the SyncNet scores between the two differ larger than a predefined threshold, and $A$ is the lower one, $A$ is determined to be in a *listening state*. **(2) Non-co-present listening dialogue**. In scenarios where speakers are not simultaneously visible, a person is classified as being in a *listening state* if the ASR result is valid and the detected SyncNet score for that person's video is below a pre-defined threshold. In both cases, the resulting listening pair is composed of the speaker's audio track and the listener's silent video track.

## 4.5 SPEAKVID-5M MULTI-TURN BRANCH

To enable multi-turn dialogue capabilities in dyadic interactive scenarios, we reserve the sequential index of multiple clips extracted from the same original video. We collect and organize two types of data for multi-turn dialogue. For a given pair of two-person dialogue clips, we define the dialogue start timestamp as $x$ and the maximum history temporal length as $T$. All clips occurring within the interval $[x - T, x]$ are considered as preceding turns of the current dialogue: **(1) Contextual multi-turn dialogue**. The corresponding ASR transcriptions of these preceding turns are aggregated to form the multi-turn dialogue context for the current response prediction. **(2) Sequential multi-turn dialogue**. We exam the temporal gap between consecutive clips. If the gap between two adjacent clips from preceding turns is less than a predefined threshold $\delta t$, we consider them part of the same continuous conversation. Every item in the sequential multi-turn dialogue contains video and audio.

## 4.6 DATA STRATIFICATION

Beyond the four categorical divisions, we also graded the data by quality. We created a high-quality supervised fine-tuning (SFT) subset (571K clips, 1368 hours) by selecting samples with a hand motion blur score above $0.5$ and face blur score above $0.7$, a DOVER score above $0.6$. We also constrain the motion score to above $2$ and filtering the ASR quality with a confidence score above $-1$. The remaining data forms the large-scale pretraining subset (7375 hours).

## 5 AUTOREGRESSIVE TALKING HUMAN GENERATION

We design a baseline method based on an autoregressive framework tailored for audio-visual dyadic human generation. As illustrated in Fig 4, we incorporate Qwen2.5-Omni Xu et al. (2025) to enable multimodal understanding of the input video and audio. Subsequently, a next-chunk prediction autoregressive model is employed to jointly generate audio and video tokens. These tokens are then used as conditioning signals for a diffusion MLP, which produces videos with enhanced realism.

### 5.1 AUTOREGRESSIVE GENERATION OF VIDEO AND AUDIO

Following Qwen2.5-Omni, we feed both the hidden states produced by the Qwen2.5-Omni thinker and the embeddings of the original audio-video inputs into the autoregressive generation head. Besides, we employ open-source 3D variational autoencoder (VAE) with a temporal stride of 4 and a spatial stride of 8 to encode the video frames to the latent space Lin et al. (2024). These latents are then divided into patches and encoded as token embeddings. For audio, we borrow audio tokenizer from CosyVoice2 Du et al. (2024) to encode raw audio into discrete tokens. We consider a latent map from the 3D-VAE and its corresponding audio tokens to constitute a single chunk. During attention computation, both audio and video tokens attend to all preceding tokens and the tokens within their current chunk. Video tokens are augmented with a combined 1D temporal and 2D spatial positional encoding. Audio tokens utilize a dual-level 1D positional encoding scheme, which encodes both the token's position within its chunk and the chunk's position in the overall sequence. The reference image is also encoded by 3D VAE, which are appended after the input audio and video and before

visual generation tokens. To mitigate error accumulation in autoregressive generation, we introduce random noise Valevski et al. (2024) to the visual tokens during training, which encourages the model to learn more robust representations and leads to improved generation quality.

## 5.2 VISUAL OPTIMIZATION

Inspired by MAR Li et al. (2024b), we also employ diffusion loss and masked-token prediction to enhance visual generation. For tokens in each generated chunk, another spatial transformer from NOVA Deng et al. (2024) takes them as input and generate a detailed visual token set-by-set. The simultaneous generation of audio-video tokens via next-chunk prediction, combined with a carefully designed frame-wise audio injection into the spatial transformer, ensures high audio-visual consistency. The visual tokens generated by the spatial transformer are temporally and spatially aligned with the latents encoded by the 3D VAE, each token corresponds to a specific patch of visual latent feature and is used as a condition signal for diffusion MLP, which performs a denoising process to generate refined visual latents that can be directly decoded by the 3D VAE. CosyVoice flow matching vocoder are used to decode the generated audio tokens into audio.

## 5.3 PROGRESSIVE TRAINING PROCESS

The model training process is divided into three stages: visual pretraining, audio-visual joint training, and high-quality dyadic dialog fine-tuning. In the **visual pretraining** stage, we utilize single-speaker data, where the ASR transcription of the target audio and textual captions (including both motion and expression captions) are used as condition signals to generate video. This stage aims to train the model's basic capability in visual content generation. In the **joint audio-visual training** stage, after filtering the audio data, we continue to use the ASR transcription and captions of the target audio as inputs, but extend the generation targets to include both video and audio modality. This stage enables the model to learn

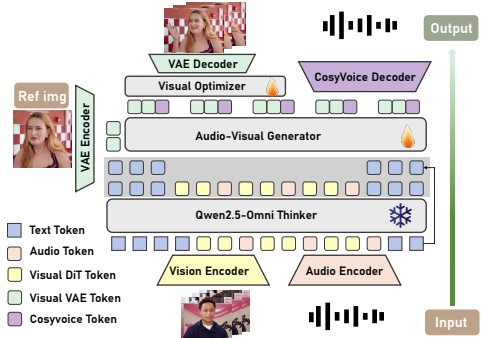

Figure 4: **Our autoregressive audio-visual generation method.**

synchronized audio-visual generation. In the **high-quality dyadic dialog fine-tuning** stage, we select premium dialogue audio-video pairs to further fine-tune the model. The goal of this stage is to enhance its multimodal understanding capabilities and its ability to generate coherent, emotionally aligned conversations. During training, the visual objective is optimized using a diffusion loss, while the audio objective is supervised with a cross-entropy loss for next-chunk prediction.

## 6 EXPERIMENTAL RESULTS AND ANALYSIS

### 6.1 THE VIDCHATBENCH BENCHMARK.

The audio-visual dyadic interactive virtual human generation involves audio-visual understanding, response generation, and the synthesis of corresponding audio and video content. It requires models to possess both understanding and generative capabilities, making it significantly more challenging than traditional generation task. To effectively evaluate the quality and appropriateness of the generated content, we constructed the **VidChatBench** benchmark, consisting of 500 representative input-output pairs with unseen speaker IDs, accompanied by a set of tailored evaluation metrics. Specifically, we assess the model performance along the following five dimensions: **(1) Video Quality**. To assess the visual fidelity of the generated videos compared to the ground truth (GT), we adopt several widely used metrics in the video and audio domains, including FID, FVD, PSNR, and SSIM. **(2) Identity Preservation**. The generated video is expected to maintain a consistent identity with the reference image throughout the sequence without temporal degradation. We employ ArcFace to extract facial features frame-by-frame from the video and compute the cosine distance between these features and the reference image. The average distance across all frames is used as the identity preservation score. **(3) Dialogue Coherence**. The generated content should exhibit semantic

| Method | Audio | Spatial | Noise | FID↓ | FVD↓ | PSNR↑ | SSIM↑ | ArcFace↑ | CLIP$_{dialog}$↑ | Sync$_{conf}$↑ | FID$_{Emotion}$↓ | SIM-o↑ |
|---|---|---|---|---|---|---|---|---|---|---|---|---|
| Conditioned | | | | 56.82 | 55.06 | 15.26 | 0.62 | 0.638 | – | – | 3.45 | – |
| Conditioned | ✓ | | | 57.03 | 55.16 | 15.31 | 0.62 | 0.630 | – | 2.063 | 3.45 | 0.65 |
| Conditioned | ✓ | ✓ | | 38.53 | 34.64 | 16.79 | 0.64 | 0.732 | – | 2.459 | 3.36 | 0.64 |
| Conditioned | ✓ | ✓ | ✓ | 34.72 | 30.43 | 17.39 | 0.65 | 0.758 | – | 2.655 | 3.23 | 0.65 |
| Dyadic | | | | 49.97 | 47.23 | 15.74 | 0.62 | 0.637 | – | – | 3.48 | – |
| Dyadic | ✓ | | | 49.86 | 36.90 | 15.63 | 0.62 | 0.635 | 0.642 | 2.239 | 3.43 | 0.64 |
| Dyadic | ✓ | ✓ | | 35.67 | 31.28 | 17.44 | 0.65 | 0.749 | 0.643 | 2.541 | 3.33 | 0.65 |
| Dyadic | ✓ | ✓ | ✓ | **32.35** | **28.82** | **17.55** | **0.66** | **0.772** | **0.643** | **2.698** | **3.22** | 0.65 |

Table 2: **Quantitative results of our baseline method on VidChatBench benchmark.** Audio means generating audio jointly. Spatial means the set-by-set prediction spatial transformer. Noise means the addition of noise in training the autoregressive audio-visual generator.

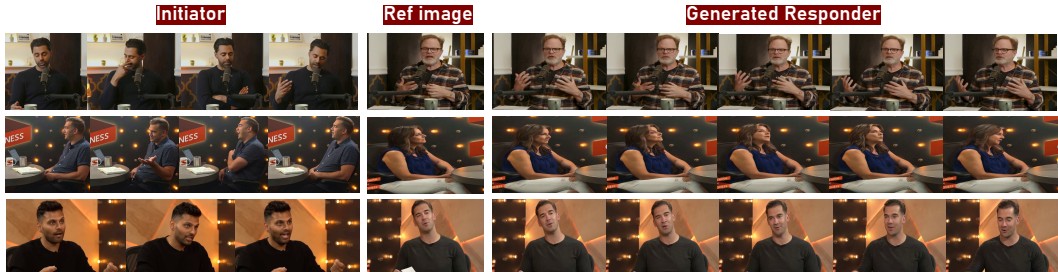

Figure 5: **Qualitative results of our dyadic generation model.** From left to right, the input video of the initiator, the reference image, and the mode's generated audio-visual response.

relevance and appropriateness with respect to the input. For each evaluation sample in our test set, we construct 5 candidate responses of varying quality, generated and ranked by a multimodal large language model Bai et al. (2025). A score is then assigned to each candidate according to its rank, using the ordered set $[0.2, 0.4, 0.6, 0.8, 1.0]$. During testing, we obtain the ASR transcription of the generated audio and compute its CLIP Radford et al. (2021) distance to each candidate response. The final dialogue coherence score is the score of the closest-matching candidate. **(4) Audio-Visual Consistency**. Synchronization between audio and lip movements is critical for the perceived realism of the generated video. We evaluate this alignment using the SyncNet confidence score, where a higher score indicates better synchronization between the generated audio and lip motion. **(5) Emotional Alignment**. In human interaction, the generated portrait should exhibit emotionally appropriate and temporally coherent expressions. We utilize Deep3DFaceRecon Deng et al. (2019b) to extract 64-dimensional expression features from both the generated and GT videos. The FID of these expression features measures the emotional alignment between the generated and GT videos. **(6) Audio Identity Preservation**. Following Chen et al. (2024), we compute the timbre similarity (SIM-o) for evaluating the timbre of generated audio. The timbre of our generated audio is controlled by the reference audio of the CosyVoice decoder.

## 6.2 Qualitative and Quantitative Results

Our primary quantitative evaluations and ablation studies are presented in Table 2, conducted on VidChatBench. We assess our model under two distinct protocols: The **Conditioned** Setting, where generation is conditioned on textual inputs, specifically the ground-truth ASR transcript and a detailed video description. The **Dyadic** Setting, where the model generates a response directly from the audiovisual input of an initiator, thereby simulating a natural dyadic interaction.

The results reveal several key insights. First, the dyadic setting significantly outperforms the conditioned setting, which we attribute to the richer, fine-grained information preserved in direct audiovisual inputs compared to abstracted textual information. Second, our joint audiovisual generation approach successfully maintains high video quality, demonstrating that incorporating audio as an additional condition does not degrade visual fidelity. Furthermore, our ablation studies validate two crucial components: (1) the spatial transformer, which yields substantial improvements in visual metrics by refining frame-level visual tokens, and (2) the training noise injection strategy, which effectively mitigates error accumulation in the autoregressive process, enhancing overall video generation quality.

## 7 Conclusion

We present SpeakerVid-5M, a large-scale, high-quality, and richly annotated dataset developed for the task of audio-visual dyadic interactive virtual human generation, which is the first dataset of its

kind. We also introduce the VidChatBench benchmark to facilitate standardized model evaluation. In addition, we provide a baseline dyadic interactive method trained on SpeakerVid-5M. We categorize SpeakerVid-5M into a large-scale pretraining subset and a curated high-quality subset for SFT. Experiments demonstrate the effectiveness of our proposed dataset and the high-quality SFT data.

# 8  ETHICS STATEMENT

The creation of our dataset, which is derived from publicly available YouTube videos, was guided by a careful consideration of the ethical dimensions of using online data, including privacy, copyright, and potential biases. This statement outlines the principles that governed our data collection, processing, and release. Speakervid-5M dataset, including all annotations and video URLs, is provided solely for scientific research and non-commercial use under the CC-BY-NC 4.0 License.

**Copyright and Terms of Service.** To respect the intellectual property of content creators and comply with YouTube's Terms of Service, we adopt a zero-resource hosting policy. We release *only* (1) a list of YouTube Video IDs/URLs, (2) timestamps, and (3) derived behavioral annotations (e.g., "waving hand"). The annotations are factual descriptions and do not contain creative expression from the original video, thus not constituting a derivative work under copyright law. We do not host, distribute, or provide tools to download raw video or audio. Access to the original content remains under the creators' full control; if a creator deletes a video or changes its privacy settings, it becomes inaccessible in our dataset. This approach is consistent with Fair Use and non-consumptive research.

**Legal Risk Acknowledgment and User Liability.** We fully respect the intellectual property and digital rights of all content creators. **1) Strict Opt-Out Policy:** If any creator wishes to retract their content, we commit to the immediate removal of the corresponding links and annotations from our dataset index upon verification of the request. **2) No Download Assistance:** We explicitly state that we do not provide, host, or distribute any raw video/audio files, nor do we offer tools or scripts to circumvent platform restrictions. **3) User Sole Responsibility:** The acquisition of raw media is the sole responsibility of the end-user. Users must independently ensure their downloading activities comply with YouTube's Terms of Service, copyright regulations, and obtain necessary consents where applicable. **4) Terms of Access and Liability:** Access to the SpeakerVid-5M annotations and indices is contingent upon the researcher's explicit agreement to our Data Use Terms. Researchers must (1) guarantee that the data will be utilized solely for non-commercial, academic research purposes, and (2) agree to assume full legal and ethical responsibility for their usage, indemnifying the video creator against any claims arising from misuse or copyright violations.

**Privacy, PII Removal, and Consent.** Given the scale of the dataset (5M clips), individual consent is infeasible. We operate under the principle of Legitimate Interest for academic research, supplemented by strict Privacy-by-Design measures: **1) Automated PII Filtering:** During our annotation process, we do not identify or label personally identifiable information. We further employ OCR (PaddleOCR Cui et al. (2025) + Qwen Yang et al. (2025)) and ASR (Whisper Radford et al. (2023)) pipelines to detect and exclude clips containing sensitive private information such as phone numbers, credit card details, or full addresses. **2) Main Speaker Focus:** Our pipeline crops the main speaker with YOLO, minimizing the inclusion of non-consenting bystanders. **3) Opt-Out Mechanism:** Beyond relying on YouTube's native removal, creators can request the removal of their specific entries from our dataset index by raising an issue in our GitHub, honoring the "Right to be Forgotten."

**Bias and Impact on Trained Models.** As our data is primarily sourced from existing online platforms, we recognize that the dataset may contain inherent societal biases and potential demographic imbalances. Our data is predominantly English (82%) and White (57%), with more demographic statistical results in the Appendix. Consequently, models trained on this data may exhibit performance disparities when processing languages or cultural gestures that are underrepresented. We encourage future research to focus on mitigating such biases to promote equitable performance across diverse demographic groups. Refer to the Appendix (A.1-5) for detailed bias and ethics analysis.

**Disclaimer.** All underlying content derived from publicly available sources (e.g., videos) remains the property of its respective rights holders; no new rights are claimed. This project is provided 'as is' without warranty. Users are solely responsible for legal compliance, and the authors disclaim all liability for any claims or damages arising from the dataset's use.

## ACKNOWLEDGEMENT

This work was supported by the STI 2030-Major Projects under Grant 2021ZD0201404 and Shenzhen Key Laboratory of New Generation Interactive Media Technology Innovation(ZDSYS20210623092001004).

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

# A APPENDIX

## A.1 DATASET STATISTICS AND DEMOGRAPHIC BREAKDOWN

Table 3: Demographic Characteristics

| Category | Subgroup | % |
|---|---|---|
| **Gender** | | |
| | Human | 62 |
| | Woman | 38 |
| **Age** | | |
| | Under 18 | 12 |
| | 18-30 | 36 |
| | 31-50 | 34 |
| | Above 50 | 18 |
| **Race** | | |
| | White | 57 |
| | Asian | 16 |
| | Black | 11 |
| | Latino | 9 |
| | Indian | 4 |
| | Mid. Eastern | 3 |
| **Region** | | |
| | US | 61 |
| | Europe | 20 |
| | China | 8 |
| | India | 5 |
| | Other | 6 |
| **Language** | | |
| | English | 82 |
| | Chinese | 9 |
| | Hindi | 4 |
| | Japanese | 2 |
| | Other | 3 |

We acknowledge that web-scraped data is susceptible to geographical, linguistic, and demographic biases, such as an overrepresentation of English/Western content. Based on available metadata and automated analysis, we provide the following statistical breakdown of our dataset with respect to gender, ethnicity, geography, age, and language: Details as follows:

The statistical results are shown in Table 3. While Western content is relatively overrepresented, the dataset still exhibits diversity in gender, age, race, region, and language. It is important to situate our work in context. The primary goal of this dataset is to establish a methodology for constructing dyadic interactive virtual human data for academic research. As an early-stage effort in this specific area, our immediate priority was to realize and study core dyadic functionalities. We anticipate that issues of bias and fairness will be progressively addressed as this research direction matures. Given the global diversity of populations and inherent imbalances in online content creation, it is exceedingly difficult for any web-scraped dataset to achieve perfect demographic parity. By providing a detailed and transparent account of our data collection and processing pipeline, we empower other researchers to adapt the methodology for specialized or more balanced datasets if needed.

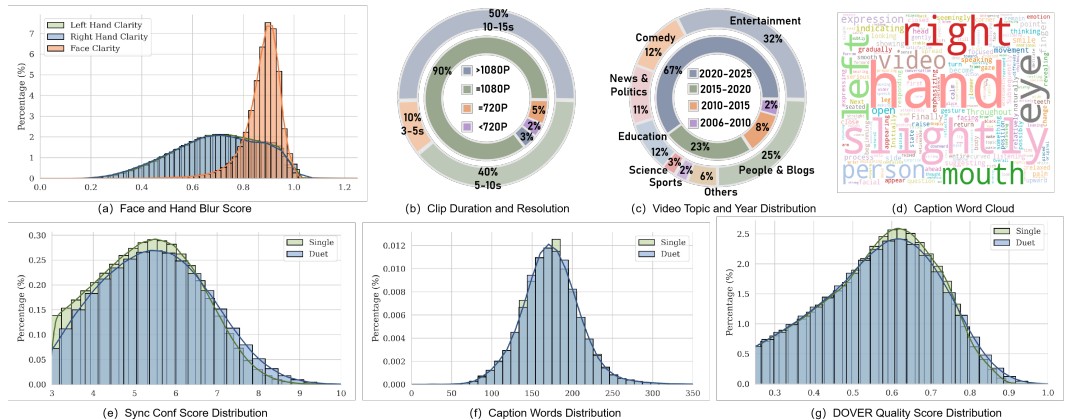

Figure 6: **Qualitative results of our dyadic generation model.** From left to right, the input video of the initiator, the reference image, and the mode's generated audio-visual response.

Figure 6 shows the statistical analysis of SpeakerVid-5M. The blur score distribution, resolution and DOVER quality distribution demonstrate the high quality of our proposed dataset. Video topic and year distribution and the caption words distribution show that the dataset exhibits diversity in monologue or conversation content. In terms of body composition, full-body, half-body, and head account for 21%, 67%, and 12%, respectively. The ratio of side view to front view is 78% and 22%.

## A.2 DETAILED ETHICS STATEMENT

We wish to clarify our data handling procedures. We do not provide direct downloads of the original videos, nor do we distribute the raw video or audio data in any form. Our dataset consists exclusively of YouTube video links and the corresponding complete annotations that we have generated.

**Terms of Use for Dataset Users.** Access to SpeakerVid-5M is gated by a license agreement that requires users to:

- Acknowledge that all right of the raw media belongs to the original creator.
- Agree to delete any raw data if the original YouTube video is taken down or made private by the creator.
- Indemnify the dataset authors against any legal claims arising from the user's use of the raw media.

**Handling Non-Consensual Data and PII.** While our data source is public, we recognize that not all subjects in YouTube videos explicitly consented to be in a dataset. To mitigate this:

1. **No Bystanders:** Our data processing pipeline (Section 3.2) specifically crops the active speaker. We discard background bystanders who did not consent to be the focus of the video.
2. **Sensitive Content Filter:** We utilize the tags form youtube to filter out content flagged as potentially harmful, violent, or sexually explicit, further reducing the risk of including non-consensual recordings (e.g., conflict footage).
3. **No Identifiable Information:** During our annotation process, we do not identify or label personally identifiable information. We further employ OCR and ASR pipelines to detect and exclude clips containing sensitive private information (e.g., credit cards, phone numbers)

**Acknowledgment of Legal Risks.** While we believe our dataset curation falls under Fair Use (as detailed in the main text), we explicitly acknowledge the legal complexities surrounding web-scraped datasets. (1) **Jurisdictional Variance:** Copyright exceptions for text and data mining (TDM) vary significantly across countries (e.g., EU vs. US vs. Asia). Users of SpeakerVid-5M must verify that

their use of the URLs and subsequent downloading of videos complies with the copyright laws of their specific jurisdiction. (2) **Platform Terms:** YouTube's Terms of Service may change. While our URL-only distribution minimizes risk, the act of downloading videos is performed by the end-user. We disclaim liability for any ToS violations committed by third-party users utilizing our index. (3) **License of Annotations:** Our annotations are released under CC-BY-NC 4.0, strictly prohibiting commercial use to further mitigate legal exposure regarding the "commercialization" of scraped data.

**Consent and Opt-Out Protocol (Subject Rights).** Since the dataset comprises over 5 million clips from public channels, obtaining individual prior consent is procedurally impossible. However, we strictly respect the agency of data subjects through the following measures: (1) **Public Source Constraint:** We only indexed videos that were publicly available at the time of collection. (2) **Dynamic Filtering:** We rely on YouTube's API. If a creator deletes a video or changes privacy settings to "Private," our dataset users will be unable to download it, effectively respecting the creator's revocation of consent. (3) **Active Opt-Out Mechanism:** We have established a dedicated communication channel (GitHub issue) for takedown requests. If a data subject or copyright holder identifies their content in our index and wishes for it to be removed, we commit to removing the corresponding annotations and URL references from the dataset within 48 hours of verification.

**Creator Rights and Privacy:** We fully respect the rights of the original content creators, who retain full control over their videos on the YouTube platform. Regarding personally identifiable information, our annotations do not introduce any personally identifiable information. Any creator can request the removal of their video's annotations from our dataset, and we are committed to modifying the dataset to comply with their needs. **Compliance with Copyright and Data Protection:** Regarding the various copyright terms and data protection regulations you mentioned, our compliance is aligned with that of the YouTube platform. Since we only provide links and do not host or distribute any raw data, access to the content is governed entirely by YouTube's Terms of Service, copyright policies, and the permissions set by the content owners. Our use of links as the sole method of access ensures we operate within these established frameworks. **Adherence to Community Standards:** The methodology we have adopted for releasing our dataset—providing only annotations and links without distributing the raw audio/video data—is entirely consistent with the established precedent set by numerous prior works in the field, most notably the VoxCeleb Nagrani et al. (2017) dataset.

**Compliance with YouTube Content License.** Our dataset construction process respects the scope of the YouTube Terms of Service and Content License. According to YouTube's license, users are prohibited from redistribution or creation of derivative works that reproduce or substitute the original content. Importantly, the public dataset we release contains no video frames, audio samples, thumbnails, metadata tied to identity, or any information from which the audiovisual content—or any person's identity—could be reconstructed. The released annotations consist solely of high-level behavioral labels (audio and video captions) and time-aligned event descriptors, which are factual observations that abstract away all expressive, visual, or acoustic elements of the underlying media. These abstract annotations do not replicate, extract, or redistribute YouTube content, nor do they enable reconstruction or serve as a substitute for the original videos. Therefore, the released dataset does not fall under the prohibited forms of copying or derivative content redistribution defined in the YouTube Content License. All annotation work was performed internally, and only non-copyrightable factual observations are released. Because the dataset includes only annotations + hyperlinks for attribution—without providing any copyrighted media—we maintain full compliance with the YouTube Content License.

**YouTube copyright and takedown framework.** YouTube's copyright policies confirm that copyright holders control copying and derivative uses of their works and provide a formal DMCA takedown mechanism and related processes for rights holders to request removal. We do not reproduce or redistribute copyrighted audiovisual content in our released artifacts; therefore, we are not engaging in the kinds of copying that the copyright regime targets. Nevertheless, if a rights holder or YouTube submits a valid removal request identifying specific video IDs in our dataset, we commit to promptly removing the corresponding annotation entries and documenting the takedown. This takedown policy and responsiveness will be explicitly stated in our README to honor creators' rights and platform procedures.

**Fair Use Consideration (U.S. Copyright Law).** Our data collection and release protocol aligns with the principles of Fair Use under U.S. copyright law (17 U.S.C. §107). Fair use is evaluated through four factors: (i) the purpose and character of the use, (ii) the nature of the copyrighted work, (iii) the

amount of copyrighted content used, and (iv) the effect on the potential market. (1) Our annotations provide a highly transformative and purely research-oriented use of publicly accessible online videos. The released dataset contains no audiovisual material, no frames, no audio waveforms, no thumbnails, no face or identity attributes, and no features from which the original content could be reconstructed. Only abstract behavioral and temporal labels—non-copyrightable factual observations—are included. (2) Although the underlying videos are creative works, our dataset does not reproduce any expressive aspect of those works. (3) We distribute zero copyrighted content, and the annotations do not capture substantive or qualitative elements of the original videos. (4) The dataset poses no market substitution risk, as it cannot replace or replicate any YouTube content and does not diminish the economic value or viewership of the original works. Hyperlinks are provided solely for attribution and do not circumvent YouTube's access controls. These factors place the dataset well within Fair Use.

**Compliance with the EU General Data Protection Regulation (GDPR).** We assessed our process under the requirements of the EU General Data Protection Regulation (GDPR), which applies only to personal data relating to identifiable individuals. (1) Anonymity of Released Artifacts: The released annotation files contain no personal data (no faces, voices, names, or biometric attributes). They consist strictly of abstract behavioral labels and timestamps. (2) Nature of Annotations: The annotations are factual, high-level descriptions that, in isolation, do not identify any natural person. (3) Data Minimization: Adhering to the principle of data minimization, we do not distribute any audiovisual content. We provide only YouTube URLs, which act as pointers to publicly available information. We do not host or process the biometric data contained within the source videos. (4) Right to be Forgotten: Since our dataset relies on video links to YouTube, if a creator removes their video (exercising their right to erasure/be forgotten), the content automatically becomes inaccessible via our dataset. (5) No Sensitive Data Transfer: As the released package contains only text-based factual annotations and public URLs without hosting biometric data, the release does not constitute a transfer of sensitive personal data under GDPR. Thus, our release strategy respects the privacy of individuals and aligns with GDPR principles.

**Compliance with General Personal Information and Privacy Regulations.** Beyond YouTube policies, Fair Use, and GDPR, we also evaluate our dataset construction under widely adopted personal information and privacy regulations (e.g., U.S. state privacy laws, OECD privacy principles, and global personal data protection frameworks). Our dataset does not contain any content that falls under these categories. Specifically: (1) No Personal Identifiers: The dataset excludes private names, contact details, or biometric records. While we include YouTube Video IDs for indexing, these identifiers point strictly to publicly available content managed by the original creators. (2) No Distributable Biometrics: We do not publish frames, audio, or feature embeddings. All biometric information remains exclusively on the hosting platform (YouTube) and is not part of our distributed artifacts. (3) Abstract Annotations: The dataset consists solely of high-level behavioral labels. These are factual descriptions of public dialogue events which, in isolation, do not reveal personal identity. (4) Public Availability Exemption: Our dataset indexes content voluntarily made public by creators. This aligns with the "publicly available information" exemptions found in regulations like CCPA, as we do not expose sensitive data not already disclosed by the owners. (5) Global Principles: Our workflow respects purpose limitation by restricting data utility to academic research and ensuring no private data is processed beyond what is necessary for indexing. By limiting the release to factual annotations and public pointers, our work respects the boundaries of personal privacy while enabling reproducible research. The dataset does not expose private or sensitive information not already publicly disclosed by the content owners.

### A.3 DISCUSSION OF POTENTIAL BIASES AND EXPLICIT WARNINGS

As a dataset derived from a large-scale, real-world platform like YouTube, it is crucial for users to understand that our dataset inherits the platform's intrinsic biases. We have not attempted to balance the dataset, and therefore, it should not be considered a representative sample of the global population. We strongly urge all researchers to conduct a thorough bias analysis before using this dataset and to be transparent about its limitations in their own work.

**Geographic and Linguistic Bias:** The data is heavily skewed towards English speakers from North America and Europe, as these demographics are over-represented on YouTube. Speakers of other languages, particularly those from the Global South, are significantly under-represented. Models trained on this dataset may perform poorly for speakers with non-standard accents or languages.

**Racial and Ethnic Bias:** The dataset likely over-represents White individuals, mirroring YouTube's content creator demographics in many popular categories. This can lead to models with higher error rates for speaker identification or poorer synthesis quality for individuals from underrepresented racial and ethnic groups. **Gender and Age Bias:** Depending on the topics crawled, there may be a skew in gender representation. Furthermore, the dataset predominantly features young adults and middle-aged individuals (the primary creator demographic), with far less data from children and the elderly.

### A.4   MANDATORY DATA USE AGREEMENT

To promote responsible AI research and mitigate legal risks, access to and use of the SpeakerVid-5M dataset is strictly governed by the following terms. By downloading, accessing, or using the dataset metadata and annotations, the researcher ("User") explicitly agrees to the following:

1. **Research-Only Limitation:** The User guarantees that the dataset, including all derived data and trained models, will be used exclusively for non-commercial, academic research purposes. Any commercial use, including but not limited to product development, commercial services, or monetization of generated content, is strictly prohibited.

2. **Assumption of Full Liability:** The User acknowledges that they are solely responsible for the acquisition of raw media content (video/audio). The User agrees to assume full legal liability for their compliance with:
   - Copyright laws and regulations in their specific jurisdiction.
   - The Terms of Service (ToS) of the source platform (YouTube).
   - Data privacy laws (e.g., GDPR, CCPA) applicable to their usage.

The use of this dataset is strictly prohibited for the following purposes: **Surveillance and Monitoring:** Any application intended for persistent or mass surveillance of individuals in public or private spaces. This includes, but is not limited to, monitoring employee conversations, tracking individuals in public areas, or any form of non-consensual biometric surveillance. **Law Enforcement and Forensic Identification:** Any use for identifying individuals in the context of criminal investigation, law enforcement, or forensic voice comparison. The inherent biases and lack of controlled recording conditions make the data unsuitable for such high-stakes decisions. **Deception or Impersonation:** Any use for creating synthetic voice or face clones ("deepfakes") or other systems intended to impersonate, deceive, or misrepresent a specific individual without their explicit, enthusiastic consent. **Discriminatory Evaluation:** Any application that infers sensitive attributes (e.g., race, ethnicity, nationality, gender, medical conditions, sexual orientation) from a person's voice to make automated decisions that could impact their access to opportunities, such as hiring, loan applications, or insurance eligibility. **Commercial Identification without Consent:** Any use in commercial products for identifying or verifying a person's identity without their explicit opt-in consent for that specific purpose.

We strongly encourage all users of this dataset to adopt the following best practices: **Report Performance Disparities:** When publishing research using this dataset, researchers are strongly encouraged to disaggregate evaluation results and report on model performance across different demographic groups (e.g., by gender, accent, or other available metadata) to expose and analyze potential biases. **Maintain Transparency:** Clearly state the use of this dataset in any resulting publications or systems, and include a discussion of its inherent limitations and biases. **Focus on Positive-Sum Applications:** We encourage the use of this dataset for applications that benefit society and advance scientific understanding, such as improving accessibility tools, developing assistive technologies, and conducting linguistic research.

### A.5   SPEAKERVID-5M DATASET LICENSE AND TERMS OF USE

1. License for Annotations and Metadata The annotations, metadata, and benchmark splits provided in the SpeakerVid-5M dataset are licensed under the Creative Commons Attribution-NonCommercial 4.0 International License (CC BY-NC 4.0).

You are free to: Share and adapt the material for non-commercial research purposes.

Under the following terms: You must give appropriate credit (cite our paper) and you may not use the material for commercial purposes.

2. Mandatory Data Use Agreement (Terms of Access) By accessing, downloading, or using the SpeakerVid-5M dataset, you (the "User") formally agree to the following terms:

A. Non-Commercial Research Only: The User guarantees that the dataset and any derived models will be used solely for non-commercial, academic research. Any commercial use, including product development or training proprietary models for commercial deployment, is strictly prohibited.

B. User Liability for Media Acquisition: The User acknowledges that the dataset authors provide only metadata and URLs. The authors do not host, distribute, or provide assistance for downloading raw video/audio data. The User assumes full legal responsibility for independently acquiring the raw media, ensuring compliance with: YouTube's Terms of Service (ToS); Copyright laws applicable in the User's jurisdiction; Data privacy regulations (e.g., GDPR).

C. Indemnification: The User agrees to indemnify, defend, and hold harmless the authors, their affiliated institutions, and funding agencies from any claims, damages, liabilities, or expenses arising from the User's use of the dataset, including but not limited to copyright infringement or violation of platform terms.

D. Respect for Creators (Opt-Out): The User agrees to delete any data associated with a specific creator if that creator requests removal or deletes their original content from the source platform.

## A.6 REPRODUCIBILITY STATEMENT

To ensure the reproducibility of our research, we provide a comprehensive account of our dataset's creation. Section 3 (Dataset Curation) offers a detailed description of our entire pipeline, including the methodologies for data sourcing, pre-processing, annotation, and filtering. Further implementation details, such as the specific prompts used for data construction and a guide to using our annotation, are provided in the Appendix. More details of our baseline method can also be found in the Appendix. The complete dataset, including the list of video URLs, our final annotations, and all source code required to replicate our data collection and processing pipeline, will be publicly available.

## A.7 IMPLEMENTATION DETAILS

During both training and inference, we standardize the frame rate to 8 FPS and resize all video frames to 480×768 resolution. The VAE compresses the input with a temporal stride of 4 and a spatial stride of 8. We treat each 4×4 feature patch as a token embedding, resulting in 360 tokens per frame in the latent space. The spatial transformer further refines each latent frame into 1440 tokens. For the audio modality, each chunk contains 12 audio tokens. To initiate generation, we introduce learnable special tokens as start-of-generation embeddings for both the audio and visual tokens. The Qwen2.5-Omni Thinker is kept frozen throughout the training process, while all other components remain trainable, resulting in a total of 0.8 billion trainable parameters. The learning rate is set to 1e-4, with warm-up and decay strategies applied. The visual pretraining and joint audio-visual training are conducted over 15 days on 128 NVIDIA L40S GPUs, with video clips in 3-7 seconds. The fine-tuning stage is carried out on 32 NVIDIA A800 GPUs over 5 days, using clips ranging from 3 to 14 seconds.

## A.8 MODEL ARCHITECTURE

**Next-chunk Prediction.** In our framework, a chunk represents the total number of tokens for one VAE latent frame and its corresponding audio. For example, at a 480x768 resolution, a single chunk consists of 360 visual tokens and 12 audio tokens. A chunk represents 4 raw video frames, corresponding to a duration of 0.5 seconds at 8 fps. The visual tokens we use are continuous features, which are derived by applying a patch embedding layer to the latent features extracted by the VAE. The next-chunk prediction refers to how our Audio-Visual Generator (an AR transformer) generates a chunk of visual and audio tokens in one forward process. The generated visual tokens are also referred to as coarse-grained visual tokens, as they contain more temporal dynamics information than spatial details. The spatial details will be further refined with the Visual Optimization module.

**Spatial Transformer.** Our Visual Optimization module is composed of two primary components: a Spatial Transformer and a Diffusion MLP. The role of the Spatial Transformer is to spatially refine the coarse-grained visual tokens for the current chunk (Generated by the Audio-Visual Generator). It takes coarse-grained visual tokens as input and produces fine-grained visual tokens as output. This refinement process is executed on a set-by-set basis. Specifically, the visual tokens for a single chunk are partitioned into multiple sets, and the Spatial Transformer processes one set at a time. Through multiple iterations, it generates the complete sequence of fine-grained visual tokens for the entire chunk. These resulting fine-grained tokens then serve as the conditioning input for the Diffusion MLP. Refer to the "Generation Process Details" part for details of set-by-set generation.

**Diffusion MLP.** The Diffusion MLP in our implementation is not a DiT, it is a lightweight 3-block MLP with adaptive LayerNorm conditioning. Concretely, the model (1) patch-embeds the noisy video latents into tokens via PatchEmbed, (2) computes a per-token condition z (visual-condition tokens generated by spatial transformer) which is augmented by a sinusoidal time embedding (get time + visual condition with simple addition of the sinusoidal time embedding and visual-condition tokens), and (3) applies stacked MPL blocks for diffusion, each of which uses an adaptive LayerNorm (AdaLayerNorm) to inject the (time + visual) condition into the residual MLP path. The block structure is an MLP sandwiched around AdaLayerNorm with a learned multiplicative gate on the MLP update. This design is intentionally simpler and much cheaper than a DiT (no self-attention), while still providing strong, per-patch conditioning via AdaLayerNorm. The conditional visual tokens for Diffusion MLP are generated by the Spatial Transformer.

**AR Generation Loop.** During inference, the generation loop proceeds as follows: First, the Audio-Visual Generator uses next-chunk prediction to output a new chunk of audio and visual tokens. The predicted visual tokens from this chunk then serve as a conditioning signal for the Visual Optimization module. This module uses the visual tokens to generate a refined, high-fidelity video latent feature. For the subsequent generation step, this refined video latent feature from the diffusion MLP is re-processed through the patch embedding layer. The resulting tokens are then appended to the sequence and used as the updated history for the next prediction by the Audio-Visual Generator.

## A.9 GENERATION PROCESS DETAILS

During inference, the generation loop proceeds as follows: First, the Audio-Visual Generator uses next-chunk prediction to output a new chunk of raw audio and visual tokens. The predicted visual tokens from this chunk then serve as a conditioning signal for the Visual Optimization module. This module uses the visual tokens condition to generate a refined, high-fidelity video latent feature. For the subsequent generation step, this refined video latent feature from the diffusion MLP is re-processed through the patch embedding layer. The resulting tokens are then appended to the sequence and used as the updated history for the next prediction by the Audio-Visual Generator. The generation process is a system of two nested loops. Taking 480x768px video at 8fps as an example.

**The Outer Loop (Chunk-by-Chunk Autoregressive Generation)** At the beginning of the generation process, with all token embeddings of the audio-visual inputs (the initiator's audio-visual stream and the responder's reference image) and thinker output, the Audio-Visual Generator (an AR Transformer) performs a single forward pass to produce the first chunk. A chunk consists of 372 tokens: 360 coarse-grained visual tokens and 12 audio tokens. These 360 coarse visual tokens are then passed to the Visual Optimization module. This module first uses a Spatial Transformer to refine them into 1,440 fine-grained visual tokens. These fine-grained tokens then condition a Diffusion MLP to generate the final, high-fidelity video latent feature for that chunk. To close the loop, this final video latent feature is re-processed via patch embedding to create new coarse-grained visual tokens. These tokens are then appended to the history and used by the Audio-Visual Generator to predict the next chunk. This constitutes the first, outer loop for next chunk prediction.

**The Inner Loop (Set-by-Set Visual Refinement)** The Visual Optimization module itself contains a second, inner loop that operates on a set-by-set basis to generate the final latent for a single chunk. A chunk of vision tokens is randomly divided into several token sets. The Spatial Transformer refines the coarse tokens to produce the first set of fine-grained visual tokens. This set conditions the Diffusion MLP to generate a partial video latent feature. Crucially, this partial latent feature is then used by the Spatial Transformer in the next iteration of the inner loop, helping it to generate

Table 4: **Comparison of the offline cascaded solution with our end-to-end baselines.**

| Method | FID ↓ | FVD ↓ | PSNR ↑ | SSIM ↑ | ArcFace ↑ | Sync$_{conf}$ ↑ | FID$_{Emotion}$ ↓ | Infer Time ↓ | Hand Quality ↑ |
|---|---|---|---|---|---|---|---|---|---|
| Qwen2.5-omni + CosyVoice + Sonic | 33.26 | 30.52 | 17.38 | 0.61 | 0.692 | 2.972 | 3.73 | 31.43 | 0.21 |
| Qwen2.5-omni + CosyVoice + Hallo3 | **28.43** | **27.65** | 17.31 | **0.69** | **0.775** | **3.324** | 4.15 | 45.82 | 0.42 |
| Ours | 32.35 | 28.82 | **17.55** | 0.66 | 0.772 | 2.698 | **3.22** | **3.17** | **0.49** |

the subsequent set of fine-grained tokens. This process repeats until all sets for the chunk have been refined. This constitutes the second, inner loop for set-by-set refinement.

In essence, the generation of a single video can be viewed as three nested stages: (1) Next-chunk prediction: The Audio-Visual Generator creates audio and coarse visual tokens. (2) Set-by-set refinement: The Spatial Transformer generates fine-grained visual tokens from the coarse ones. (3) Denoising: The Diffusion MLP generates the final video latent feature from the fine-grained tokens. Crucially, these stages are not a simple cascade but are executed in a nested, cyclical manner. The prediction of chunk N depends on the final latent feature from chunk 0 to chunk N-1. Similarly, the refinement of set i within a chunk depends on the latent feature generated from set 0 to set i-1. We will provide technical details and diagrams in the supplementary material.

## A.10 Comparison with Cascade Pipeline with Diffusion models

Our primary goal is to explore an end-to-end solution for the audio-visual dyadic interactive virtual human generation task by developing an integrated autoregressive (AR) framework that unifies audio-visual understanding and generation. While a cascaded approach appears viable, it suffers from several inherent deficiencies, including information loss, computational redundancy, and error accumulation. During the intermediate text generation step, a wealth of non-verbal information from the initiator, such as their environment, facial expressions, and prosody, is irretrievably lost. Similarly, crucial aspects of the intended response, including tone, emotion, and corresponding gestures, are not captured in the text representation. Subsequently, the Text-to-Speech (TTS) process, while converting text to audio, can introduce further discrepancies. What's more, although proceeding in visual quality, the current audio-driven method still mainly focuses on head-and-shoulder and struggles with meaningful actions. This results in a final digital human that, despite potentially high visual quality and accurate lip-sync, is semantically and emotionally detached from the original interaction context.

Given the rapid advancements in diffusion-based video foundation models, it is unsurprising that cascaded solutions currently lead in certain metrics. However, their fundamental limitations severely restrict their applicability in truly interactive scenarios. Therefore, the exploration of end-to-end alternatives remains both necessary and beneficial. As no open-source cascaded systems for this task are currently available, we constructed two offline cascaded baselines for comparison, both build with a visual language model (VLM) for text response, a Text-to-Speech (TTS) model for response audio acquisition, and an audio-driven human animation model for video generation. For the VLM component, we chose qwen2.5-omni, which aligns with our own method and has powerful video/audio understanding capabilities. For TTS, we used the robust Cosyvoice model. For the final audio-driven human video synthesis, we select Hallo3 Cui et al. (2024) (~10B parameters) and Sonic Ji et al. (2025) (~1.5B parameters).

To better illustrate the inherent performance disparities between our end-to-end method and the cascaded pipelines, we propose two additional evaluation metrics. (1) HQ (Hand Quality): This metric assesses the quality of hands as the geometric mean of Hand Clarity (HC) and Hand Fluctuation (HF), formulated as $\sqrt{HC \cdot HF}$. This design penalizes both blurry moving hands and unnaturally static hands. HC is the Laplacian sharpness of the hand region detected by DW-Pose, while HF is calculated as 1 - IoU of hand bounding boxes in consecutive frames. (2) Single-Frame Inference Time (Infer Time): We report the average time required to generate a single frame. This is included because timely interaction is a critical factor in dyadic generation, and the architectural choice between autoregressive (AR) and diffusion models has a significant impact on this aspect.

We must acknowledge that our method is at an inherent disadvantage in this comparison due to the lack of a comparable, large-scale pre-trained AR foundation model. Furthermore, our audio-visual generation module has only 0.8B parameters, a fraction of the size of the cascaded pipelines (Hallo3+Cosyvoice at ~10B; Sonic+Cosyvoice at ~2B).

The experimental results are presented in Table 4. Despite being disadvantaged in terms of visual priors and parameter count, our method outperforms both cascaded baselines on the emotion and

hand quality metrics, and exhibits a significant advantage in the single-frame inference time. The superiority in emotion and hand quality stems from our end-to-end architecture, which preserves high-level semantic and affective information that is lost during the intermediate text conversion in the cascaded approach. In the cascade solution, the audio-driven diffusion model lacks awareness of both the context of the initiator and the appropriate actions to take. This results in inaccurate emotional expressions and near-static hand movements. Additionally, beyond the preservation of high-level semantic information, the inference time is another critical factor that influenced our preference for an end-to-end architecture. A stark contrast in inference efficiency is observed between the two solutions. Our end-to-end method generates a single frame in just 3.17 seconds. In contrast, the Hallo3-based pipeline requires a substantial 45.82 seconds per frame. Even the lower-parameter Sonic-based pipeline remains prohibitively slow, at 31.43 seconds per frame. This significant disparity in inference time renders the direct application of large, cascaded diffusion models for talking human generation computationally infeasible for the dyadic interactive scenario. Conversely, the inherent efficiency of our end-to-end autoregressive model provides a clear advantage, making it a far more viable pathway for optimization into a truly usable, interactive digital human. Regarding video quality metrics, our method is outperformed by the much larger Hallo3-based pipeline but surpasses the Sonic-based one, which has a comparable parameter count. The cascaded approaches benefit from their powerful, specialized base models, whereas our method's strong performance is bolstered by our diverse full-body and half-body training data.

A.11   ANNOTATION FILE USAGE

In this section, we provide a detailed explanation of the annotation files in our SpeakerVid-5M dataset to promote the application. The basic annotation file serves as a central repository for essential metadata pertaining to each clip. This includes: Source video ID and URL: Enabling direct access to the original video content. Clip timestamps: The precise start and end times of the clip within the source video. Human bounding box: Spatial localization of the primary subject within the clip's frames. Video resolution: The dimensions of the video content. SyncNet confidence score: A metric assessing the lip-sync quality. DOVER score: A quality score of overall video clip quality. Clarity score: An indicator of visual clarity. Speaker ID: Unique identification based on audio for the speaker present in the clip. These comprehensive annotations are designed to empower users to directly retrieve the original videos from YouTube and efficiently extract high-quality single-person audiovisual clips using straightforward FFmpeg commands. Furthermore, we provide open-source code that facilitates the intelligent linking of these individual clips, thereby enabling the convenient construction of two-person dialogues and more complex multi-turn dialogue datasets.

**The `l_score` Annotation.** The `l_score` annotation provides quantitative clarity metrics for salient regions within each video, namely the left hand, right hand, and face. For each region, we compute Laplacian-based scores on a frame-by-frame basis. These are provided as both absolute scores, which are comparable across the entire dataset, and relative scores, normalized within each clip. Furthermore, the frame-wise absolute scores are aggregated to produce a single, holistic clip-level score. This hierarchical annotation is designed for two primary use cases: the fine-grained, frame-level scores serve as dynamic conditioning signals for generative models, enabling more precise control over motion fidelity. In contrast, the clip-level scores offer an effective mechanism for dataset curation, allowing researchers to easily filter for high-clarity training samples to enhance model robustness.

**The `Caption` Annotation.** The `Caption` annotation file provides a rich set of clip-level semantic labels, automatically generated using large multimodal model (qwen2.5 vl). These annotations offer a structured, multi-faceted description of each video's content, enabling detailed analysis and control. The specific labels are as follows:

- **Video Quality:**
  - **Clarity:** A binary label indicating the absence or presence of significant motion blur or artifacts.
  - **Camera Dynamics:** Classification of camera motion (e.g., static, panning, zooming) and shot framing (e.g., close-up, medium shot).
- **Subject & Scene Composition:**
  - **Subject Count:** The number of individuals detected in the frame.

- **Framing View:** A label indicating if the subject is framed in a full-body or upper-body view.
- **Head Pose:** The estimated orientation of the subject's head (e.g., frontal, side view).
- **Scene Entities:** A list of recognized objects and persons present.

- **Behavioral & Action Details:**
  - **Speech Activity:** A binary label indicating active speech.
  - **Motion Status:** A binary label classifying the primary subject as either static or in motion.
  - **Movement Intensity:** A categorical rating of the subject's motion intensity (e.g., low, medium, high).
  - **Holistic Action Summary:** A high-level textual description of the subject's overall actions.
  - **Fine-grained Action Description:** A detailed textual account of specific, nuanced actions performed by the subject.
  - **Facial Expression Summary:** A textual description of the subject's primary facial expression.

Collectively, these annotations furnish a structured representation of the video content. This enables both fine-grained conditioning for generative tasks and provides a robust framework for the objective evaluation of synthesized visual attributes and behaviors.

**The `Scene` Annotation.** The `Scene` annotation provides the temporal metadata required for managing and assembling video clips from the original source footage. This annotation is generated through a two-stage process. First, an initial scene detection algorithm segments the source videos into coherent shots, recording their start/end timestamps and frame indices. Subsequently, these segments are further partitioned to enforce a maximum duration of 14 seconds per clip. The final annotation file catalogs each processed clip with a unique identifier and its precise temporal boundaries (start and end times). This structured metadata is crucial for downstream applications, enabling the seamless concatenation of clips for long-form video synthesis and providing a flexible framework for tasks involving multi-turn dialogue or extended narrative generation.

**The `Speaker` Annotation.** `Speaker` annotation is performed on the original audio of video recordings to provide detailed speaker diarization results. This comprehensive annotation process involves several key components. Initially, raw diarization results are generated, capturing the precise start and end times for each detected speaker turn, along with their respective assigned speaker IDs. These results provide a foundational temporal map of all spoken segments. For applications focused on dyadic dialogue scenarios, the raw results undergo a filtering process. This step specifically identifies and prioritizes the two primary speakers (labeled as Speaker A and Speaker B). A critical criterion for this filtering is that these two speakers must collectively account for at least 80% of the total speech duration within the video. Furthermore, in the context of dyadic data filtering, only the conversational exchanges occurring exclusively between Speaker A and Speaker B are retained. This ensures the selection of highly relevant dialogue for subsequent model training and analysis. The final stage yields a cleaned and refined list of speaker IDs and their corresponding start and end times for each segment. This provides clear and accurate speaker attribution of the video, ensuring high-quality data for downstream tasks.

**The `ASR` Annotation.** `ASR` annotation is performed on unified single-person audiovisual clips, which are meticulously derived from praevia video and audio processing. For example, in dual-person co-present scenes, the segmentation process spatially divides the video into two distinct regions, each corresponding to an individual. Concurrently, the associated audio is temporally segmented to align with the speech of each respective person. To achieve these single-person audiovisual clips—where the video prominently features only one person and the audio contains solely that person's speech—we integrate YOLO for visual analysis with speaker diarization for audio segmentation. Upon obtaining these refined clips, automatic speech recognition is then applied to their corresponding audio segments. The ASR output encompasses several key components. Transcribed speech text: The verbatim textual representation of the spoken content. Confidence score: A metric indicating the reliability and accuracy of the transcription. Speech compression ratio: The ratio between the original speech duration and its compressed form, relevant for efficiency analysis.

No-speech probability: The likelihood that a given audio segment contains no discernible speech. Language information: Identification of the language spoken within the clip.

## A.12    VISUALIZATION OF PRETRAIN AND FINETUNE MODEL

As illustrated in Figure 7, the model resulting from our initial pre-training phase exhibits pronounced motion blur, particularly in the face and hands. These artifacts are significantly exacerbated during rapid movements, degrading perceptual quality. To address this, we curated a high-quality subset by filtering the training data based on facial and hand clarity. The figure demonstrates that fine-tuning on this refined data yields substantial improvements in visual fidelity, markedly reducing blur and enhancing detail in regions critical for human perception. To enable this data refinement, we compute blur scores for both the face and hands in each frame using a Laplacian-based method. Recognizing their broader utility, we release these scores as part of our public dataset annotations. The value of such explicit quality signals is substantiated by prior work; for instance, CyberHost demonstrated that conditioning a model on hand blur scores can significantly enhance the clarity and fidelity of synthesized hand motions.

## A.13    PROMPT USED IN ANNOTATION

Tables 5 and 6 present the caption prompts used for Qwen-VL-2.5, which serve as effective guidance for generating high-quality responses during captioning.

## A.14    FAILED CASE AND ANALYSIS

(1) Fine-grained Hand Dynamics: Prevailing talking-head datasets are predominantly constrained to rudimentary hand motions, such as resting, crossing, or waving. Consequently, the synthesis of more complex and expressive actions, including intricate gesturing or human-object interactions, remains a formidable challenge in digital human generation. This difficulty is compounded by the fact that textual descriptions typically lack the requisite granularity to supervise these fine motor skills, thereby hindering a model's ability to learn rich hand dynamics. To address this limitation, we provide comprehensive annotations for each video clip, encompassing hand keypoints, bounding boxes, and a novel hand-blur score. These annotations furnish precise supervisory signals, enabling more effective control and rigorous evaluation of synthesized hand motion quality.

(2) Synthesis of Occluded or Out-of-Frame Body Parts: A core challenge in this domain is the synthesis of body parts that are occluded or absent in the reference image, for instance, rendering a newly raised arm or animating eyes from a closed to an open state. This task necessitates models endowed with robust generative priors and a comprehensive understanding of human anatomy and plausible motion. The challenge is often exacerbated by data bias; many existing datasets are heavily skewed towards front-facing, tightly-cropped portraits, which lack diverse viewpoints and full-body context. To counteract this, our dataset incorporates a wide array of perspectives, including full-body, upper-body, and profile views. This provides a more balanced distribution of human body configurations, thereby empowering models to generalize beyond the visible input and mitigate in-painting artifacts.

(3) Temporal Coherence in Long-Form Video Generation: Generating temporally coherent, long-form video presents two primary obstacles. First, the scarcity of high-quality, long-duration training data fundamentally limits model performance. Our dataset addresses this by providing a large corpus of high-fidelity clips (3–14 seconds) that can be seamlessly concatenated into extended sequences (*e.g.*, 20 seconds to 3 minutes). Second, autoregressive models are prone to error accumulation, which degrades generation quality over time. To mitigate this, we introduce a simple yet effective noise injection strategy during training. This technique enhances the model's temporal stability, ensuring consistent quality over extended generative horizons.

## A.15    LIMITATION AND FUTURE WORK

While SpeakerVid-5M represents a significant contribution to the field of interactive virtual humans, we still discuss the current scope of this work and highlight several promising avenues for further development. First, our baseline model is presented as a preliminary benchmark to validate the

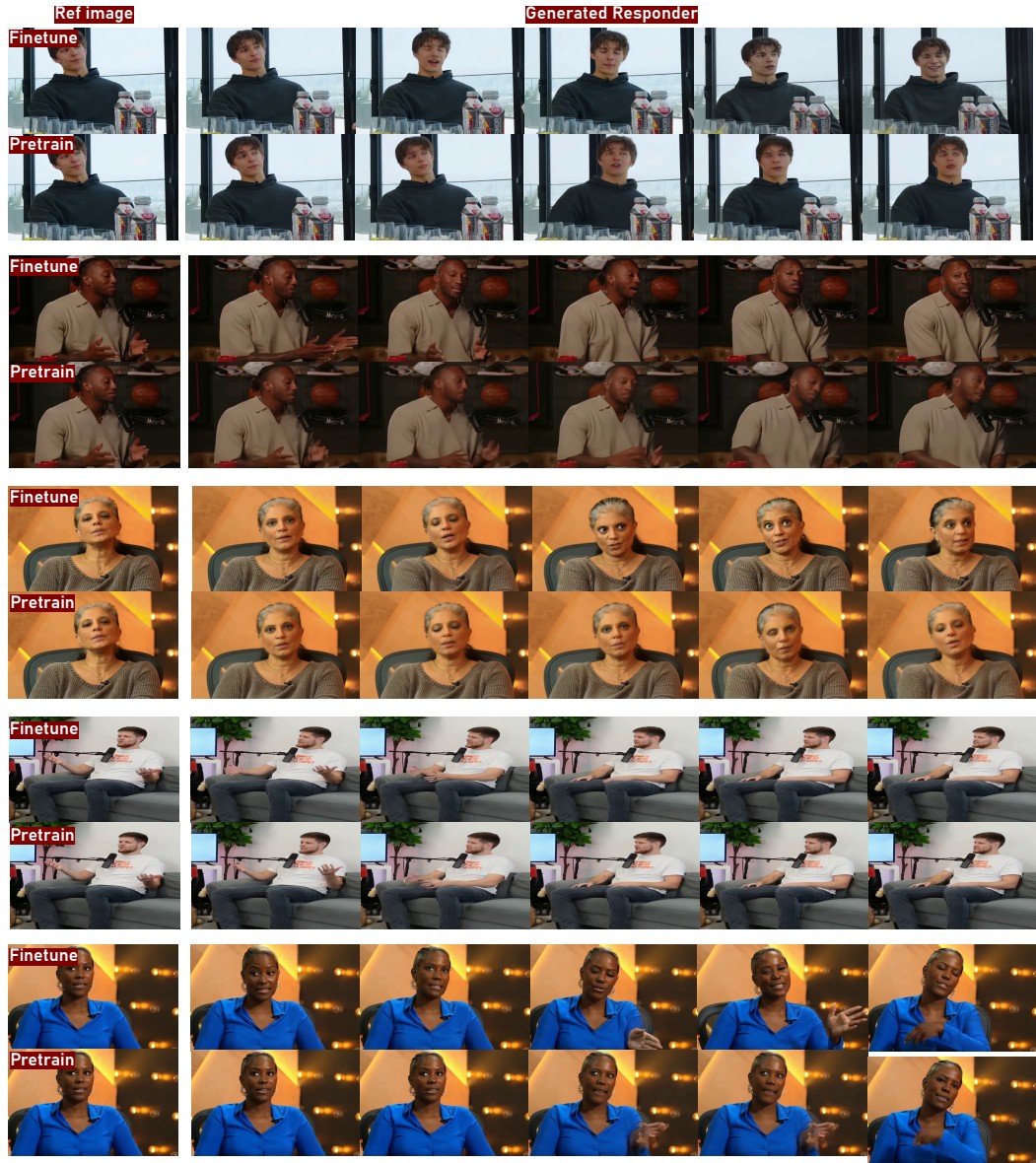

Figure 7: **Impact of finetuning on generation quality.** A comparison between the model after the pretraining phase and after finetuning on our high-quality subset.

SpeakerVid-5M dataset. Its performance was constrained by limited computational resources and does not represent the state-of-the-art. We anticipate that future work using more advanced architectures and large-scale training will unlock the dataset's full potential and yield higher-fidelity results. Second, the scope of interaction within SpeakerVid-5M is focused on single-person and two-person (dyadic) scenarios. The dataset does not currently capture the complex dynamics of multi-party conversations, such as group turn-taking. Extending the data collection to include these multi-person interactions is valuable for future research.

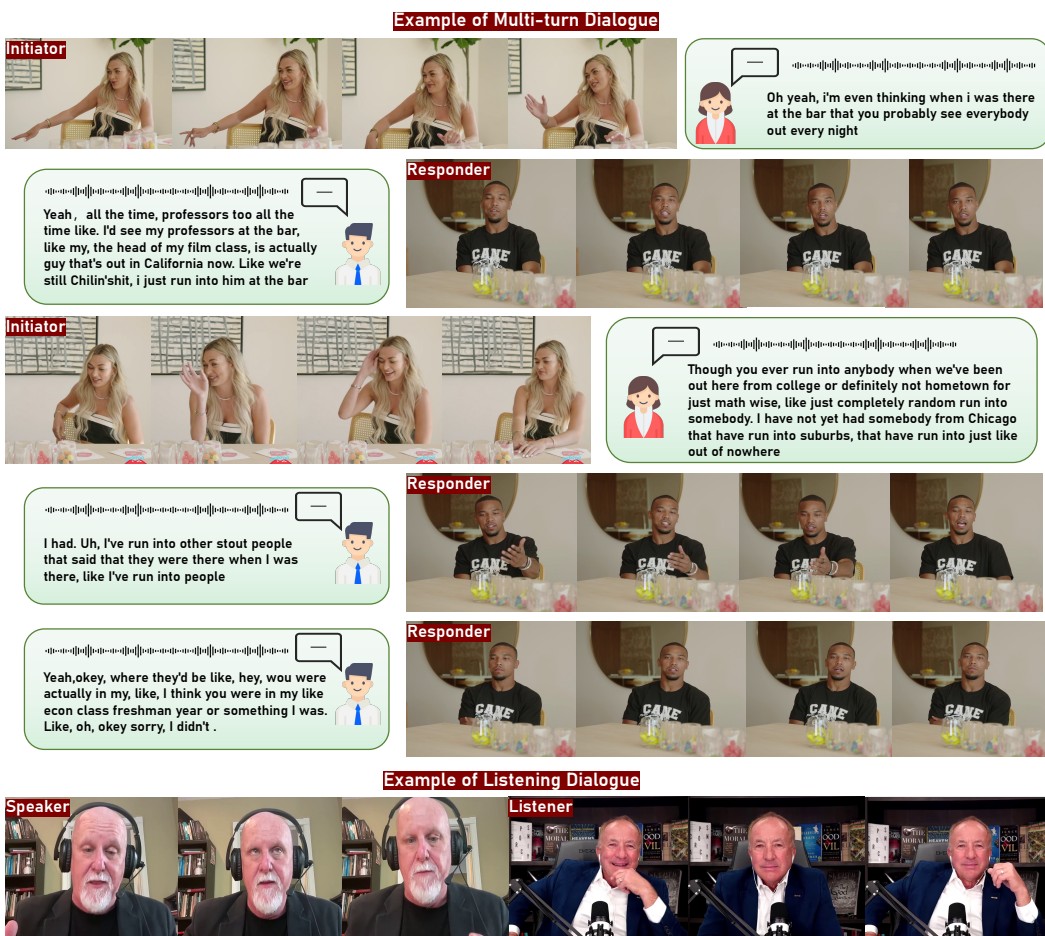

Figure 8: **Visualization of multi-turn dialogue and listening scenarios.** Top (Multi-turn Dialogue): We showcase a sequence of conversational turns between an initiator and a responder. By preserving temporal context, our dataset facilitates the training of models capable of coherent, long-form conversations. Bottom (Listening Case): A speaker is paired with a non-speaking listener.

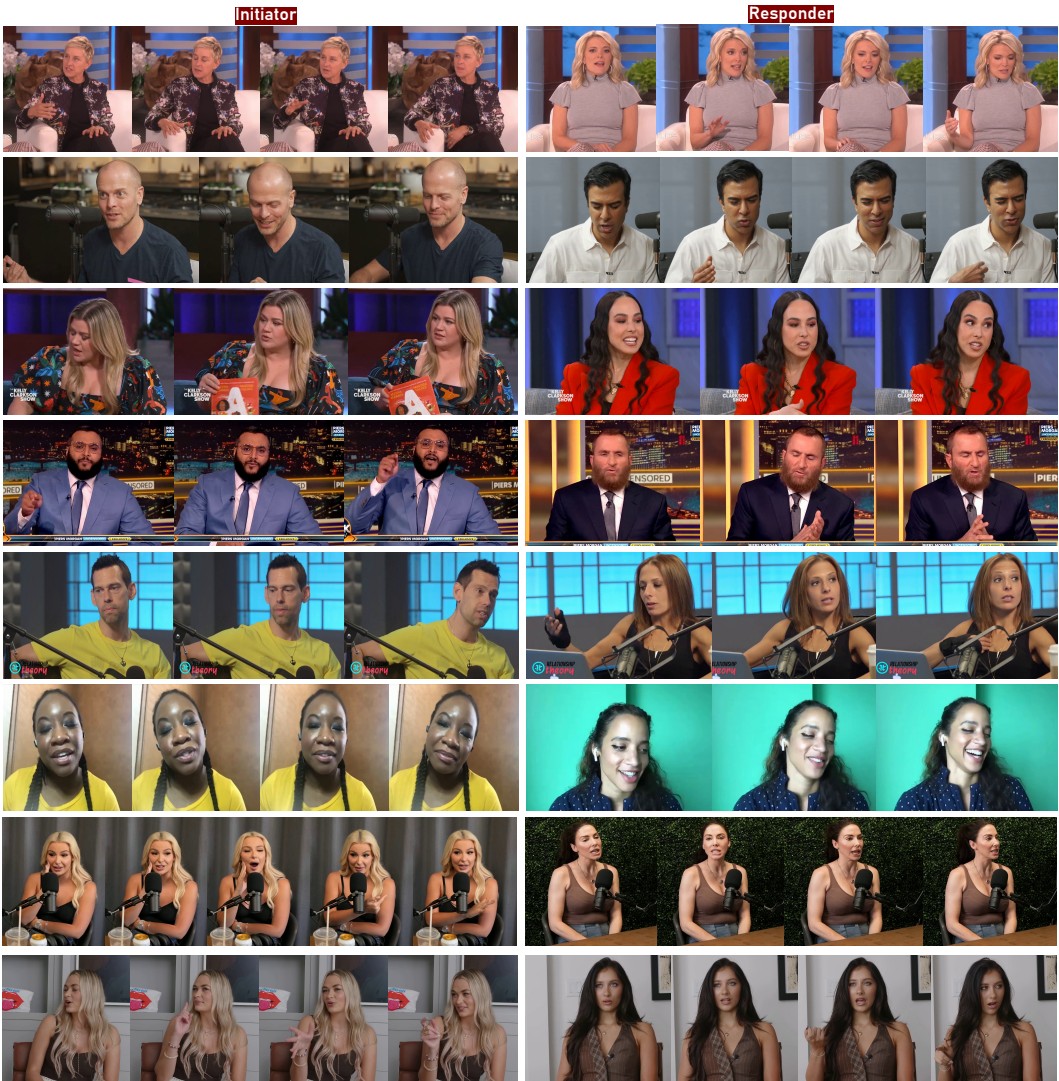

Figure 9: **Diverse examples of dyadic pairs from SpeakerVid-5M.** This figure highlights the diversity of subjects, environments, and interaction styles captured in our dataset.

Table 5: Prompt Used in Video Annotation

# Your Role: Video Annotation Expert

## Objective: As a professional annotator, your task is to evaluate and label video content based on clarity, camera motion, human presence and activity, and semantic understanding. Your annotations must be structured and follow a predefined JSON format.

## Annotation Guidelines For each input video, assess and annotate the following attributes:

1. **Visual Clarity:** Assess whether the primary subject in the video is clearly visible and distinguishable. Options: `"Yes"` or `"No"`.

2. **Camera Motion:** Identify the type(s) of camera movement observed. Choose one or more from the following: `["Static Camera", "Dynamic Shooting", "Still Shot", "Left-to-Right Pan", "Right-to-Left Pan", "Zoom In", "Zoom Out"]`.

3. **Human Motion Presence:** Determine whether the person in the video exhibits obvious and recognizable physical movement. Options: `"Yes"` or `"No"`.

4. **Motion Intensity Level:** Rate the level of physical activity on a scale from 1 (very minimal motion) to 5 (extremely intense movement). Aim to differentiate clearly across the full range rather than concentrating ratings in the middle.

5. **Entity List:** List all visually prominent entities in the video in descending order of saliency, e.g., `["man", "teddy bear", "river", "traffic sign", "apple"]`.

6. **Speech Presence:** Identify whether the person is clearly speaking in the video. Options: `"Yes"` or `"No"`.

7. **Observed Actions:** List the specific actions shown in the video, such as `["running", "dancing", "eating", "talking", "singing", "presenting"]`.

8. **Number of People:** Count the number of distinct people visible in the video.

9. **Upper-Body Only:** Indicate whether only the upper half of the person is visible (if the legs or lower body appear, select "No"). Options: `"Yes"` or `"No"`.

10. **Facing Direction:** Specify the subject's orientation relative to the camera. Options: `"Front"`, `"Side"`, or `"Back"`.

## Output Format: Return the annotation results in the following JSON structure:

1. **Clarity:** `"Yes"`

2. **Camera Motion:** `["Zoom In", "Left-to-Right Pan"]`

3. **Human Motion Presence:** `"Yes"`

4. **Motion Intensity Level:** `[3]`

5. **Entity List:** `["man", "dog", "apple"]`

6. **Speech Presence:** `"Yes"`

7. **Observed Actions:** `["talking"]`

8. **Number of People:** 1

9. **Upper-Body Only:** `"Yes"`

10. **Facing Direction:** `"Side"`

**Note:** Please ignore any subtitles or on-screen text when performing the annotation. Focus solely on the video's visual and auditory content.

Table 6: Prompt Used in Video Motion Intensity Level

# Your Role: Human Motion Expert

## Annotation Guidelines: As a human motion expert, assess the intensity of movement in the video by focusing on the amplitude and frequency of body movements: 1. Movement Amplitude: Whether the person frequently makes large body movements like arm swings, body turns, etc. 2. Movement Frequency: Whether the person repeats similar movements frequently, such as nodding their head, gesturing, etc. Score range from 1 to 5.

1. The person is nearly stationary, with only minimal head or small hand movements.
2. The person makes occasional small gestures or minor body adjustments.
3. The person uses moderate gestures with occasional body adjustments, such as slight forward leans.
4. The person uses frequent gestures, with larger body movements and more frequent body shifts.
5. The person's movements are frequent and large, with extensive use of hand gestures and body shifts, indicating high intensity.

# Your Role: Audience Member

## Annotation Guidelines: As an audience member, assess the movement intensity of the speaker by focusing on their emotional expression and the resulting body movements: 1. Emotional Fluctuations: Whether the speaker shows large emotional fluctuations in their speech, and whether it is accompanied by body language. 2. Audience Reactions: Whether the speaker adjusts their body in response to audience reactions (e.g., applause, nodding). Score ranges from 1 to 5.

1. The speaker speaks in a calm, even tone with minimal body movement.
2. The speaker has slight emotional fluctuations, with occasional small gestures or head movements.
3. The speaker shows moderate emotional fluctuations and occasional body movements like hand gestures or body shifts.
4. The speaker shows significant emotional fluctuations, with frequent body movements, gestures, and emotional intensity.
5. The speaker displays intense emotional fluctuations, with frequent and large gestures and body movements, indicating high intensity.

# Your Role: Labeling Expert

## Annotation Guidelines: As a labeling expert, assess the intensity of movement by considering the body language and interaction frequency: 1. Gesture Usage: Whether the person frequently uses hand gestures or body movements to emphasize their speech. 2. Interaction Frequency: Whether the person interacts frequently with others, especially through body language responses (e.g., nodding, smiling). Score ranges from 1 to 5.

1. The person makes little to no gestures or body movements and interacts minimally.
2. The person occasionally uses gestures or makes small body movements, with limited interaction.
3. The person frequently uses gestures and makes moderate body adjustments, with moderate interaction with others.
4. The person uses hand gestures frequently, with larger body movements and frequent interaction.
5. The person has frequent and strong body movements, with highly frequent interactions and large gestures, indicating high intensity.

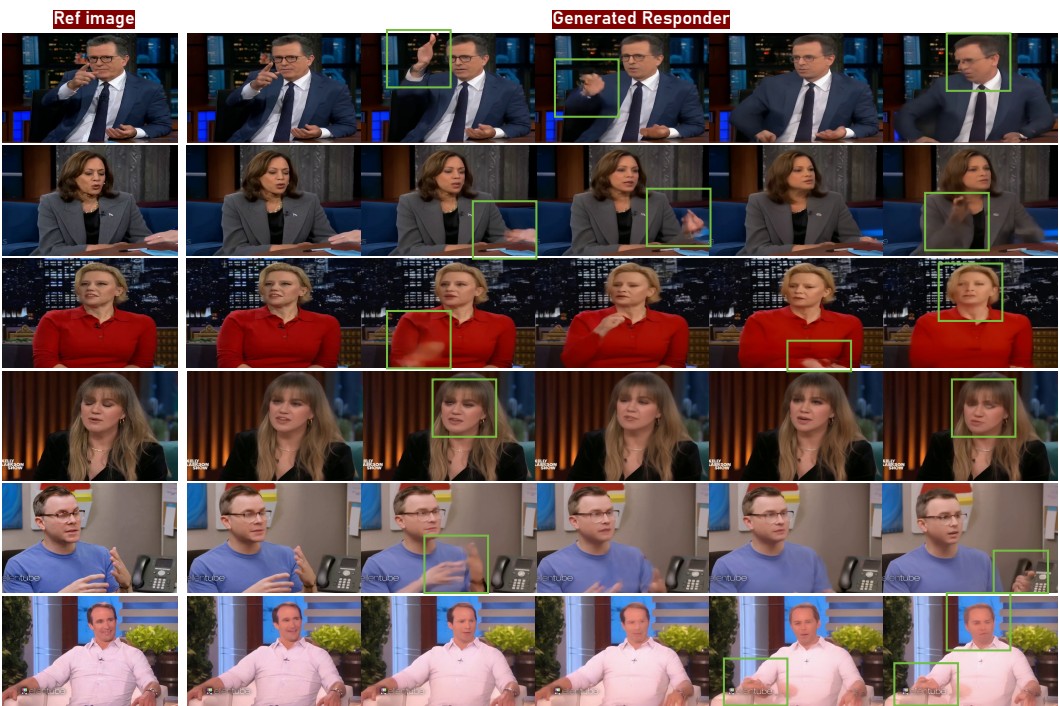

Figure 10: **Analysis of failure cases.** These include generating facial artifacts or unnatural distortions and struggling with severe motion blur during rapid head or hand movements.

