# OpenReview forum: "SpeakerVid-5M: A Large-Scale High-Quality Dataset for Audio-Visual Dyadic Interactive Human Generation"
_ICLR.cc/2026/Conference — ICLR 2026 Poster_

### Official Review · Reviewer_8FnX · 2025-10-24

**Soundness:** 3
**Presentation:** 3
**Contribution:** 2
**Rating:** 6
**Confidence:** 2

**Summary:**

This paper presents SpeakerVid-5M, a large-scale audio-visual speaker understanding dataset containing 5 million labeled video clips paired with aligned face and speech data. The authors aim to enable multi-modal representation learning for speaker identification, speech-to-face generation, and lip-synchronization tasks. The dataset is collected from online sources (e.g., YouTube) using an automated pipeline that detects talking faces, aligns speech segments, and filters content for language and quality.

**Strengths:**

1. SpeakerVid-5M is one of the largest open-source datasets in the speaker understanding domain, with diverse speakers, recording conditions, and languages, making it valuable for pretraining multimodal encoders.

2. By integrating speaker verification, lip-reading, and voice–face retrieval within one dataset, it provides a practical testbed for representation transfer and multi-task learning.

3. The use of automatic speech-face alignment and quality filtering (lip-sync thresholding, blur detection) demonstrates thoughtful engineering and scalability.

**Weaknesses:**

1. The core contribution is the dataset’s size, not a new method, model, or theoretical insight. It extends existing pipelines rather than rethinking them.

2. There is little quantitative characterization of dataset diversity, no breakdown by gender, age, language, recording environment, or cultural representation.

3. Large-scale scraping of online videos raises questions about consent, copyright, and potential bias amplification. The ethical section is superficial and lacks compliance discussion (e.g., GDPR, US fair-use).

**Questions:**

1. Beyond scale, what distinguishes SpeakerVid-5M from VoxCeleb2 + LRS3? Are there new annotation dimensions (e.g., speaker emotion, interaction type, recording context) that justify the “5M” contribution?

2. Have authors quantified the rate of misaligned or noisy samples? What proportion of clips have accurate lip audio synchronization?

3. Could you report statistics on gender, age, ethnicity, and geographic diversity? How do you mitigate bias from the over-representation of Western/English speakers?

4. Are the improvements in Table 4 solely due to dataset size, or did you modify architecture or hyperparameters? A same-architecture ablation on VoxCeleb vs. SpeakerVid would help isolate the effect.

**Details Of Ethics Concerns:**

1. The dataset is collected by large-scale crawling of online videos (e.g., YouTube) without clear evidence of speaker consent or data-usage rights. There is no mention of how personally identifiable information (faces, voices) is handled or whether subjects can request removal, potentially violating privacy laws and data protection regulations such as GDPR.

2. The paper claims to use “publicly available sources,” but it is unclear whether the dataset respects copyright terms, platform terms of service, and content-owner permissions. The large-scale scraping of audiovisual data may breach website crawling policies or content licenses.

3. No demographic breakdown (gender, ethnicity, geography, or age) is provided. Given the web-based nature of data collection, there is a high likelihood of English-language and Western-centric bias, which could propagate discrimination in downstream speaker identification or face-generation applications.

---

> ### Author Response · Authors · 2025-11-19
> **Response to Reviewer 8FnX (Part 1/4)**
>
> We are grateful for your valuable suggestions for improving our manuscript. After careful consideration of the points you raised, we would like to provide the following clarifications.
>
> ## **Response 1: Core Contribution Clarification**
> Regarding the core contribution of our paper, we wish to clarify that our focus extends significantly beyond the dataset's scale.
>
> First and foremost, our dataset is the first large-scale dataset designed for the task of audio-visual dyadic interactive virtual human generation. By explicitly defining the roles of an initiator and a responder, we provide usable large-scale data for the community to tackle this new challenge. Beyond single-turn exchanges, we also provide multi-turn dialogue data to support contextual conversational scenarios. This dyadic, conversational structure represents a fundamental distinction from existing talking-head datasets.
>
> Secondly, we address a major limitation of traditional talking-head corpora, which are often confined to frontal, head-and-shoulders views, focusing primarily on lip-sync and facial expressions, and lacking overall dynamism. In contrast, our dataset is more diverse, encompassing full-body and half-body, frontal and side views, and is rich with expressive body language and hand gestures.
>
> Finally, the versatility of our dataset is embodied in its four distinct branches, each designed to advance a critical area of digital human research:
> - **Single branch:** the largest talking-head dataset to date that features full-body data and detailed hand motion.
> - **Dialogue branch:** the first resource designed to support end-to-end, audio-visual dyadic interactive generation.
> - **Listening branch:** provides crucial data on listener states, essential for agents that can not only speak but also listen actively.
> - **Multi-turn branch:** extends the dialogue paradigm to more complex, context-aware scenarios.
>
> In summary, while the scale of our dataset is a significant component, it is only part of our contribution. Our primary contribution lies in proposing a new and valuable paradigm (audio-visual dyadic interactive generation) for the future of digital human generation research and providing the first large-scale, empirical foundation to facilitate research in this direction. This forward-looking vision, substantiated by the data required to pursue it, constitutes the core of our work.
>
> ## **Response 2: Difference with Talking-head Generation Datasets**
> First, we would like to clarify that the "5M" in our name refers to the dataset's scale, indicating that it comprises nearly 5 million synchronized audio-visual clips. However, the core contributions of SpeakerVid-5M extend far beyond its scale and are distinguished from traditional talking-head datasets in several fundamental ways:
>
> **A Novel Task Formulation:** SpeakerVid is the first large-scale dataset designed for dyadic interactive virtual human generation. For this task, each data instance includes the audio-visual streams for both an initiator and a responder. The objective is to generate the responder's complete audio-visual output given the initiator's input. This represents a significant paradigm shift from conventional audio-driven tasks that focus on a single subject.
>
> **Task Versatility:** SpeakerVid is organized into four distinct branches (single, dialogue, listening, multi-turn), each targeting a critical aspect of conversational digital human generation. This multi-faceted structure makes its application scope significantly broader than any existing dataset.
>
> **Superiority in the Traditional Domain:** While our **single branch** shares a similar format with existing talking-head datasets, it surpasses them in both scale and visual clarity, establishing a new state-of-the-art resource even for the conventional talking-head task.
>
> **Enhanced Data Diversity and Dynamism:** Unlike traditional datasets that are often limited to frontal, head-and-shoulders views, our dataset incorporates full-body, half-body, frontal, and side-view perspectives. This results in greater diversity and richer motion dynamics.

---

> ### Author Response · Authors · 2025-11-19
> **Response to Reviewer 8FnX (Part 2/4)**
>
> ## **Response 3: Dataset Diversity and Demographic Breakdown**
> We acknowledge that web-scraped data is susceptible to geographical, linguistic, and demographic biases, such as an overrepresentation of English/Western content. Based on available metadata and automated analysis, we provide the following statistical breakdown of our dataset with respect to gender, ethnicity, geography, age, and language:
> Details as follows:
>
> **Table: Demographic Characteristics**
>  | Category   | Distribution                                                             |
>  | ---------- | ------------------------------------------------------------------------ |
>  | **Gender** | Male 62%, Female 38%                                                     |
>  | **Age**    | <18 12%, 18–30 36%, 31–50 34%, > 50 18%                         |
>  | **Race**   | White 57%, Asian 16%, Black 11%, Latino 9%, Indian 4%, Middle Eastern 3% |
>  | **Region** | US 61%, Europe 20%, China 8%, India 5%, Other 6%                                  |
>  | **Language** | English 82%, Chinese 9%, Hindi 4%, Japanese 2%, Other 3%                                  |
>
> While Western content is relatively overrepresented, the dataset still exhibits diversity in gender, age, race, region, and language.
> It is important to situate our work in context. The primary goal of this dataset is to establish a methodology for constructing dyadic interactive virtual human data for academic research. As an early-stage effort in this specific area, our immediate priority was to realize and study core dyadic functionalities. We anticipate that issues of bias and fairness will be progressively addressed as this research direction matures. Given the global diversity of populations and inherent imbalances in online content creation, it is exceedingly difficult for any web-scraped dataset to achieve perfect demographic parity.
>
> By providing a detailed and transparent account of our data collection and processing pipeline, we empower other researchers to adapt the methodology for specialized or more balanced datasets if needed.
>
> To promote responsible use, we include the following in our appendix and the dataset release:
> - A statistical analysis of the dataset diversity and demographic composition.
> - A discussion of potential biases and guidance on risks for downstream applications.
> - Recommended use restrictions prohibiting automated law enforcement, surveillance, and commercial identification systems, while encouraging researchers to report performance disparities across demographic groups.
>
> ## **Response 4: Lip Audio Synchronization Clarification**
> We thank the reviewer for raising this important point. Lip-audio synchronization is indeed a critical metric for evaluating speaker-related datasets. Figure 6 in the appendix provides a detailed report of the SyncNet confidence score distribution across our dataset.
>
> Our data curation process includes a stringent filtering step: the final dataset only contains clips with a SyncNet confidence score greater than **3.0**, which corresponds to clear synchronization between the lips and the audio. This filtered dataset totals **8,700 hours**. Furthermore, over 50% of clips surpass a SyncNet confidence score of **5.5**, indicating a very high degree of synchronization. Therefore, we ensure that the dataset contains only clips with well-aligned lip and audio signals.
>
> ## **Response 5: Table 4 Clarification**
> We would first like to clarify a point regarding the table reference. Table 4 in our manuscript refers to the prompt examples provided in the appendix, not to ablation or comparison experiments.
>
> We believe you may be referring to Table 2 in the main text. If so, we wish to emphasize that the improvement demonstrated is not merely from dataset scale, but from a fundamental shift in dataset construction. Traditional talking-head datasets, such as VoxCeleb, exclusively contain single-person speaking data. They lack the dyadic interaction component—the paired initiator-responder audio-visual sequences—which is the cornerstone of our work.
>
> Consequently, these existing datasets are unsuitable for the task of audio-visual dyadic interactive virtual human generation. Our contribution is the first large-scale dataset specifically designed to enable the research of audio-visual dyadic interaction human generation.

---

> ### Author Response · Authors · 2025-11-19
> **Response to Reviewer 8FnX (Part 3/4)**
>
> ## **Response 6: Ethics Concern**
>
> As detailed in our Ethics Statement in the appendix, we wish to clarify our data handling procedures. We do not provide direct downloads of the original videos, nor do we distribute the raw video or audio data in any form. Our dataset consists exclusively of YouTube video links and the corresponding complete annotations that we have generated.
> **Creator Rights and Privacy:**
> We fully respect the rights of the original content creators, who retain full control over their videos on the YouTube platform. Regarding personally identifiable information, our annotations do not introduce any personally identifiable information. Any creator can request the removal of their video's annotations from our dataset, and we are committed to modifying the dataset to comply with their needs.
> **Compliance with Copyright and Data Protection:**
> Regarding the various copyright terms and data protection regulations you mentioned, our compliance is aligned with that of the YouTube platform. Since we only provide links and do not host or distribute any raw data, access to the content is governed entirely by YouTube's Terms of Service, copyright policies, and the permissions set by the content owners. Our use of links as the sole method of access ensures we operate within these established frameworks.
> **Adherence to Community Standards:**
> The methodology we have adopted for releasing our dataset—providing only annotations and links without distributing the raw audio/video data—is entirely consistent with the established precedent set by numerous prior works in the field, most notably the VoxCeleb dataset.
>
> ### **Compliance with YouTube Content License:**
> Our dataset construction process respects the scope of the YouTube Terms of Service and Content License. According to YouTube’s license, users are prohibited from redistribution or creation of derivative works that reproduce or substitute the original content. Importantly, the public dataset we release contains no video frames, audio samples, thumbnails, metadata tied to identity, or any information from which the audiovisual content—or any person’s identity—could be reconstructed. The released annotations consist solely of high-level behavioral labels (audio and video captions) and time-aligned event descriptors, which are factual observations that abstract away all expressive, visual, or acoustic elements of the underlying media.
>
> These abstract annotations do not replicate, extract, or redistribute YouTube content, nor do they enable reconstruction or serve as a substitute for the original videos. Therefore, the released dataset does not fall under the prohibited forms of copying or derivative content redistribution defined in the YouTube Content License. All annotation work was performed internally, and only non-copyrightable factual observations are released. Because the dataset includes only annotations + hyperlinks for attribution—without providing any copyrighted media—we maintain full compliance with the YouTube Content License.
>
> ### **YouTube copyright and takedown framework:**
> YouTube’s copyright policies confirm that copyright holders control copying and derivative uses of their works and provide a formal DMCA takedown mechanism and related processes for rights holders to request removal. We do not reproduce or redistribute copyrighted audiovisual content in our released artifacts; therefore, we are not engaging in the kinds of copying that the copyright regime targets. Nevertheless, if a rights holder or YouTube submits a valid removal request identifying specific video IDs in our dataset, we commit to promptly removing the corresponding annotation entries and documenting the takedown. This takedown policy and responsiveness will be explicitly stated in our README to honor creators’ rights and platform procedures.

---

> ### Author Response · Authors · 2025-11-19
> **Response to Reviewer 8FnX (Part 4/4)**
>
> ### **Fair Use Consideration (U.S. Copyright Law)**
> Our data collection and release protocol aligns with the principles of Fair Use under U.S. copyright law (17 U.S.C. §107). Fair use is evaluated through four factors: (i) the purpose and character of the use, (ii) the nature of the copyrighted work, (iii) the amount of copyrighted content used, and (iv) the effect on the potential market.
>
> 1. Our annotations provide a highly transformative and purely research-oriented use of publicly accessible online videos. The released dataset contains no audiovisual material, no frames, no audio waveforms, no thumbnails, no face or identity attributes, and no features from which the original content could be reconstructed. Only abstract behavioral and temporal labels—non-copyrightable factual observations—are included.
> 2. Although the underlying videos are creative works, our dataset does not reproduce any expressive aspect of those works.
> 3. We distribute zero copyrighted content, and the annotations do not capture substantive or qualitative elements of the original videos.
> 4. The dataset poses no market substitution risk, as it cannot replace or replicate any YouTube content and does not diminish the economic value or viewership of the original works. Hyperlinks are provided solely for attribution and do not circumvent YouTube’s access controls.
>
> These factors place the dataset well within Fair Use.
>
> ### **Compliance with the EU General Data Protection Regulation (GDPR)**
> We assessed our process under the requirements of the EU General Data Protection Regulation (GDPR), which applies only to personal data relating to identifiable individuals.
> 1. **Anonymity of Released Artifacts:** The released annotation files contain no personal data (no faces, voices, names, or biometric attributes). They consist strictly of abstract behavioral labels and timestamps.
> 2. **Nature of Annotations:** The annotations are factual, high-level descriptions that, in isolation, do not identify any natural person.
> 3. **Data Minimization:** Adhering to the principle of data minimization, we do not distribute any audiovisual content. We provide only YouTube URLs, which act as pointers to publicly available information. We do not host or process the biometric data contained within the source videos.
> 4. **Right to be Forgotten:** Since our dataset relies on video links to YouTube, if a creator removes their video (exercising their right to erasure/be forgotten), the content automatically becomes inaccessible via our dataset.
> 5. **No Sensitive Data Transfer:** As the released package contains only text-based factual annotations and public URLs without hosting biometric data, the release does not constitute a transfer of sensitive personal data under GDPR.
>
> Thus, our release strategy respects the privacy of individuals and aligns with GDPR principles.
>
> ### **Compliance with General Personal Information and Privacy Regulations**
> Beyond YouTube policies, Fair Use, and GDPR, we also evaluate our dataset construction under widely adopted personal information and privacy regulations (e.g., U.S. state privacy laws, OECD privacy principles, and global personal data protection frameworks).
> Our dataset does not contain any content that falls under these categories. Specifically:
>
> 1. **No Personal Identifiers:** The dataset excludes private names, contact details, or biometric records. While we include YouTube Video IDs for indexing, these identifiers point strictly to publicly available content managed by the original creators.
> 2. **No Distributable Biometrics:** We do not publish frames, audio, or feature embeddings. All biometric information remains exclusively on the hosting platform (YouTube) and is not part of our distributed artifacts.
> 3. **Abstract Annotations:** The dataset consists solely of high-level behavioral labels. These are factual descriptions of public dialogue events which, in isolation, do not reveal personal identity.
> 4. **Public Availability Exemption:** Our dataset indexes content voluntarily made public by creators. This aligns with the "publicly available information" exemptions found in regulations like CCPA, as we do not expose sensitive data not already disclosed by the owners.
> 5. **Global Principles:** Our workflow respects purpose limitation by restricting data utility to academic research and ensuring no private data is processed beyond what is necessary for indexing.
>
> By limiting the release to factual annotations and public pointers, our work respects the boundaries of personal privacy while enabling reproducible research. The dataset does not expose private or sensitive information not already publicly disclosed by the content owners.

---

### Official Review · Reviewer_Ey3j · 2025-10-31

**Soundness:** 3
**Presentation:** 3
**Contribution:** 3
**Rating:** 8
**Confidence:** 4

**Summary:**

This paper introduces a new dataset and a new benchmark, accompanied by a baseline model.
The dataset, SpeakerVid-5M, is a large-scale, high-quality collection specifically designed for audio-visual dyadic interactive virtual human generation. It contains over 8K hours of human portrait video clips.
The dataset is richly annotated with multi-modal information, including structured textual captions, human skeletal sequences (DWpose), ASR transcripts, and blur scores. It is further organized into four interaction types (dialogue, single, listening, and multi-turn branches) and divided into two tiers: a large-scale pre-training subset and a high-quality Supervised Fine-Tuning (SFT) subset.
The authors also propose an autoregressive (AR)-based video chat baseline model trained on this dataset and introduce VidChatBench, a dedicated benchmark with tailored metrics for evaluating performance on the complex task of audio-visual dyadic generation, which requires both multi-modal understanding and generation capabilities.
The dataset and processing code are planned to be publicly released.

**Strengths:**

1. The paper addresses a critical and timely problem. The research community's focus is clearly shifting from passive, audio-driven "talking heads" to proactive, interactive digital humans. The authors correctly identify that the single greatest barrier to open-source academic research in this area is the lack of a large-scale, high-quality dataset specifically capturing dyadic audio-visual interactions. This contribution directly unblocks this important future direction.
2. The dataset is a significant contribution, and its value is strongly reinforced by the rigorous and transparent data curation pipeline. The authors provide excellent clarity (particularly in Figure 2) on each methodical step, which builds trust in the data's fidelity. The use of a modern, multi-model suite for filtering and annotation is a major strength. For example, employing DOVER for video quality assessment, SyncNet to ensure tight audio-visual synchronization, and a powerful VLM like Qwen2.5-VL to generate rich, structured annotations (like motion, expression, and pose) goes far beyond simple ASR transcripts and ensures the data is high-quality and multi-faceted.
3. The proposed model architecture makes sense to me. The baseline is built on Qwen2.5-Omni. The 'thinker' to understand the initiator's multimodal input, and a joint Audio-Visual Generator that acts as a 'renderer' to produce the audio and video response. This end-to-end approach is a modern paradigm, and providing this baseline (along with the VidChatBench) gives the community a strong starting point and clearly demonstrates the dataset's immediate utility.

**Weaknesses:**

1. Limited Data Domain and Generality: The data sources, while high-quality, are heavily skewed towards formal or semi-formal scenarios (interviews, news, seminars, debates). The dataset appears to lack more casual, "in-the-wild" interaction styles, such as personal vlogs, movie/TV drama scenes, or general user-generated content. This "domain bias" might limit the ability of models trained on this data to generalize to the full spectrum of human interaction needed for a truly "general purpose" digital human.

2. Ambiguous Baseline Model Architecture: The description of the autoregressive baseline in Section 5 is too high-level, leaving several critical implementation details unclear. This makes the model difficult to reproduce or fairly assess.

"Diffusion MLP": This component is a black box. What is its architecture? Is it a simple MLP, or a more complex structure like a DiT? How are the time-step and visual token conditions precisely integrated?

"AR Generation Loop": This is the most confusing part. The model is autoregressive, but it also uses a diffusion process to "refine" latents. For the next generation step, what exactly is fed back into the AR model? Is it the initial, raw tokens from the AR generator, or the refined latents after the diffusion/denoising step? .

"Next-chunk prediction": What is the temporal size of a "chunk"? Are the visual tokens continuous or discrete token?

3. Missing SOTA Modular Baseline Comparison: The paper rightly claims no single end-to-end model exists for this task, but it fails to compare against the most obvious and powerful SOTA alternative: a modular pipeline. A proper evaluation would benchmark the proposed model against a baseline that 1) uses an LLM/VLM to generate a text response, 2) uses a TTS model for speech, and 3) uses a state-of-the-art speech-to-video model (like Hallo3, etc.) to synthesize the video. Without this comparison, it's impossible to know if the proposed end-to-end method is actually superior to (or even competitive with) what's already possible by combining existing SOTA components.

**Questions:**

Modular Baseline Comparison: Could the authors provide results for a strong modular baseline on VidChatBench?  See W3

---

> ### Author Response · Authors · 2025-11-19
> **Response to Reviewer Ey3j (Part 1/3)**
>
> We would like to express our sincere gratitude for your insightful feedback and encouraging evaluation. We deeply appreciate your efforts in helping us to further strengthen this manuscript. Having carefully considered your valuable suggestions, we have summarized our responses below.
>
> ## **Response 1: Data Domain and Generality**
> We sincerely appreciate your valuable suggestion about the data domain and the generality. Your idea is highly forward-looking and aligns with a primary challenge we aim to address in our future work. However, the curation of clean dyadic dialogue (containing both an initiator and a responder), let alone multi-turn sequences, from highly naturalistic sources like films and television dramas is fraught with difficulty. The primary reasons are as follows:
>
> 1. **Scene Complexity and Data Purity**: Films and TV drama frequently feature multiple characters with numerous entries and exits. This results in a very low proportion of pure two-person dialogue scenes, requiring massive filtering efforts to yield a small amount of usable data. Moreover, due to the visual and acoustic complexity, existing data processing pipelines struggle to maintain high accuracy compared to more formal settings like interviews.
> 2. **Sequence Fragmentation**: While scenically rich, data curated from films and general user-generated content is often fragmented. It is challenging to extract complete multi-turn conversations, with most segments capturing only a single conversational turn and thus lacking deep contextual history.
> 3. **Complex Cinematography**: Film production involves frequent and dynamic camera cuts. It is common for the shot to change even within a single sentence, which makes it extremely difficult to acquire complete, uninterrupted dialogue segments.
> 4. **Copyright Restrictions**: Film and television data are subject to more stringent copyright issues, which complicate any effort to create an open-source dataset.
>
> Given the increased complexity and lower data curation efficiency, we strategically focused our current data collection on scenarios such as interviews, news broadcasts, seminars, and debates. These contexts allow us to more reliably acquire the complete dyadic and multi-turn dialogue data that our task requires.
>
> For our future work, we plan to process high-quality, pre-segmented dialogue clips from film and television that are available on online platforms. Although these clips are shorter, they typically feature stable character IDs and a strong dialogue focus, which we anticipate will significantly improve the data yield from these more naturalistic sources.
>
> ## **Response 2: Model Architecture Explanation**
> We sincerely apologize for the confusion caused by the brevity of the description in our methods section. Due to page limitations, a more detailed breakdown was not feasible in the main paper. We provide clarification below and include a comprehensive description in the revised appendix (Section A.3: Model Architecture ).
>
> ### **Next-chunk Prediction**
> In our framework, a chunk refers to the collection of tokens representing one VAE latent frame together with its corresponding audio.
> For example, at a resolution of 480×768, a single chunk contains 360 visual tokens and 12 audio tokens. Each chunk corresponds to 4 raw video frames, corresponding to a duration of 0.5 seconds at 8 fps.
> The visual tokens are continuous features obtained by applying a patch embedding layer to the latent features produced by the VAE. Next-chunk prediction refers to the process in which our Audio-Visual Generator (an autoregressive transformer) generates both the visual and audio tokens for the next chunk in a single forward pass. The visual tokens produced at this stage are coarse-grained—they encode more temporal dynamics but less spatial detail—and their spatial fidelity is subsequently enhanced by the Visual Optimization module.
>
> ### **Spatial Transformer**
> Our Visual Optimization module consists of two main components: a Spatial Transformer and a Diffusion MLP.
> The Spatial Transformer spatially refines the coarse-grained visual tokens produced by the Audio-Visual Generator. It takes these coarse tokens as input and outputs fine-grained visual tokens.
> This refinement is performed in a set-by-set manner: the visual tokens of a chunk are partitioned into multiple sets, and the Spatial Transformer processes one set at a time. Across these steps, it produces the complete sequence of fine-grained visual tokens for the entire chunk. These fine-grained tokens then serve as conditioning input for the Diffusion MLP. Additional details of the set-by-set generation can be found in the “Generation Process Details” section in the following response.

---

> ### Author Response · Authors · 2025-11-19
> **Response to Reviewer Ey3j (Part 2/3)**
>
> ### **Diffusion MLP**
> Our Diffusion MLP is architecturally distinct from DiT-based designs. It is a compact 3-block MLP equipped with adaptive LayerNorm conditioning. The time-step and visual token conditions are integrated via broadcasting and elementwise addition of their embeddings. Specifically:
> 1. We first patch-embed the noisy video latents using a PatchEmbed layer to obtain per-patch tokens.
> 2. For each token, we compute a conditioning vector z (the visual-condition tokens produced by the Spatial Transformer), which is augmented via elementwise addition with a sinusoidal time embedding.
> 3. The diffusion process is then performed by stacked MLP blocks, where each block applies an AdaLayerNorm layer to inject the (time + visual) condition into the residual MLP pathway.
>
> The block structure is an MLP sandwiched around AdaLayerNorm with a learned multiplicative gate on the MLP update.
> This architecture is intentionally simpler than DiT, yet still provides strong patch-level conditioning via AdaLayerNorm. The conditioning visual tokens used by the Diffusion MLP are generated by the Spatial Transformer.
>
> ### **AR Generation Loop**
> During inference, the autoregressive generation loop proceeds as follows:
> 1. **Next-chunk prediction**: The Audio-Visual Generator outputs a new chunk of audio and coarse visual tokens.
> 2. **Visual refinement**: These coarse visual tokens serve as conditioning inputs to the Visual Optimization module, which produces a refined, high-fidelity video latent representation.
> 3. **Feedback for the next step**: The refined latent is patch-embedded again into tokens, which are appended to the history sequence and used as context for the next-step prediction of the Audio-Visual Generator.
>
> This loop ensures that each newly generated chunk benefits from both temporal continuity (via AR modeling) and spatial fidelity (via diffusion-based refinement).
>
> ## **Response 3: Generation Process Details**
> The generation process can be understood as a system of two nested loops. Let's use the 480x768 resolution at 8fps as an example.
>
> ### **The Outer Loop (Chunk-by-Chunk Autoregressive Generation)**
> At the beginning of the generation process, with all token embeddings of the audio-visual inputs (the initiator's audio-visual stream and the responder's reference image) and thinker output, the Audio-Visual Generator (an AR Transformer) performs a single forward pass to produce the first chunk. A chunk consists of 372 tokens: 360 coarse-grained visual tokens and 12 audio tokens.
>
> The 360 coarse visual tokens are then processed by the Visual Optimization module. The Spatial Transformer refines them into 1,440 fine-grained visual tokens, which condition a Diffusion MLP to generate the final high-fidelity video latent feature for the chunk. This latent feature is then patch-embedded again to produce new coarse-grained visual tokens, which are appended to the token sequence and used to predict the next chunk. This constitutes the first, outer loop with next-chunk prediction.
>
> ### **The Inner Loop (Set-by-Set Visual Refinement)**
> Within the Visual Optimization module, an inner loop refines each chunk in a set-by-set manner. A chunk of visual tokens is partitioned into multiple token sets. The Spatial Transformer refines the first set of coarse tokens into fine-grained tokens, which then condition the Diffusion MLP to generate a partial video latent feature. Crucially, this partial latent feature is then used by the Spatial Transformer in the next iteration of the inner loop, helping it to generate the subsequent set of fine-grained tokens. The set-specific operation is implemented via masking. This process repeats until all sets in the chunk are refined. This constitutes the second, inner loop for set-by-set refinement.
>
> In essence, the generation of a single video can be viewed as three nested stages:
> 1. **Next-chunk prediction**: The Audio-Visual Generator creates audio and coarse visual tokens.
> 2. **Set-by-set refinement**: The Spatial Transformer generates fine-grained visual tokens from the coarse ones.
> 3. **Denoising**: The Diffusion MLP generates the final video latent feature from the visual fine-grained tokens.
>
> Crucially, these stages are not a simple cascade but are executed in a nested, cyclical manner. The prediction of chunk N depends on the final latent feature from chunk 0 to chunk N-1. Similarly, the refinement of set i within a chunk depends on the latent feature generated from set 0 to set i-1. All technical details and diagrams have been added to the revised appendix.

---

> ### Author Response · Authors · 2025-11-19
> **Response to Reviewer Ey3j (Part 3/3)**
>
> ## **Response 4: Comparison with Cascade Pipeline with Diffusion Models**
>
> We would like to further clarify the motivation behind our proposed end-to-end baseline model. Our primary goal is to explore an end-to-end solution for the audio-visual dyadic interactive virtual human generation task by developing an integrated autoregressive (AR) framework that unifies audio-visual understanding and generation.
>
> Although a cascaded approach seems viable, it suffers from inherent limitations like information loss, computational redundancy, and error accumulation. For example, during the intermediate text generation step, rich non-verbal cues from the initiator—including environmental context, facial expressions, and prosody—cannot be fully preserved. Similarly, crucial aspects of the intended response, such as tone, emotion, and corresponding gestures, are not captured. Subsequent Text-to-Speech (TTS) conversion can introduce further discrepancies, and existing audio-driven methods are largely limited to the head-and-shoulder, struggling to generate meaningful actions. This results in a final digital human that, despite potentially high visual quality and lip-sync accuracy, is semantically and emotionally detached from the original interaction.
> Given the rapid advancement of diffusion-based video foundation models, it is unsurprising that cascaded solutions currently lead in certain metrics. However, their fundamental limitations constrain their applicability in truly interactive scenarios. Therefore, the exploration of end-to-end alternatives remains both necessary and beneficial.
>
> Since no open-source cascaded systems are publicly available, we constructed two offline cascaded baselines for experimental comparison, each using a visual language model (VLM) for text responses, a TTS model for audio, and an audio-driven animation model for video generation.
>
> To better illustrate the inherent performance disparities between our end-to-end method and the cascaded pipelines, we propose two additional evaluation metrics:
> 1. **HQ (Hand Quality)**: This metric assesses the quality of hands as the geometric mean of Hand Clarity (HC) and Hand Fluctuation (HF), formulated as $\sqrt{\text{HC} \cdot \text{HF}}$. This design penalizes both blurry moving hands and unnaturally static hands. HC is the Laplacian sharpness of the hand region detected by DW-Pose, while HF is calculated as 1 - IoU of hand bounding boxes in consecutive frames.
> 2. **Single-Frame Inference Time (Infer Time)**: We report the average time required to generate a single frame.
>
> | Method | FID ↓ | FVD ↓ | PSNR ↑ | SSIM ↑ | ArcFace ↑ | Sync_conf ↑ | FID_Emotion ↓ | Infer Time ↓ | Hand Quality ↑ |
> |--------|-------|-------|--------|--------|-----------|-------------|---------------|--------------|----------------|
> | Qwen2.5-omni + CosyVoice + Sonic | 33.26 | 30.52 | 17.38 | 0.61 | 0.692 | 2.972 | 3.73 | 31.43 | 0.21 |
> | Qwen2.5-omni + CosyVoice + Hallo3 | **28.43** | **27.65** | 17.31 | **0.69** | **0.775** | **3.324** | 4.15 | 45.82 | 0.42 |
> | Ours | 32.35 | 28.82 | **17.55** | 0.66 | 0.772 | 2.698 | **3.22** | **3.17** | **0.49** |
>
> We would like to clarify that our method is at a disadvantage due to the lack of a comparable large-scale pre-trained AR foundation model and our model's smaller parameter count (0.8B for the audio-visual generation module) compared to the cascaded pipelines(Hallo3+Cosyvoice at ~10B; Sonic+Cosyvoice at ~2B).
>
> As shown in the table above, despite these disadvantages, our method surpasses both cascaded baselines on emotion and hand quality metrics and demonstrates a significant advantage in inference time. This superiority stems from our end-to-end architecture, which preserves high-level semantic and affective information lost during the intermediate text conversion in the cascaded approach. The cascaded diffusion models lack awareness of the initiator's context and actions, leading to inaccurate emotional expressions and nearly static hands.
>
> In terms of inference efficiency, our end-to-end method generates a single frame in just 3.17 seconds. In contrast, the Hallo3-based pipeline requires 45.82 seconds, and even the smaller Sonic-based pipeline takes 31.43 seconds. This significant disparity renders large cascaded diffusion models less practical for interactive scenarios. Conversely, the inherent efficiency of our autoregressive model presents a much more viable path toward a truly usable, interactive digital human. Regarding video quality metrics, our method is slightly outperformed by the much larger Hallo3-based pipeline but surpasses the comparably-sized Sonic-based one.
>
> We sincerely hope our experimental results have addressed your concerns and, in doing so, offer a modest but useful insight regarding the advantages of the end-to-end approach and the choice of effective technical paradigms for the audio-visual dyadic interactive virtual human generation task.

---

> > ### Comment · Reviewer_Ey3j · 2025-11-26
> >
> > Thanks for the author's thoughtful response. I would like to maintain my positive evaluation of this work.

---

> > > ### Author Response · Authors · 2025-11-28
> > > **Thank you**
> > >
> > > We sincerely thank you for your continued engagement and for maintaining a positive assessment of our work. We remain fully dedicated to clarifying any remaining points and would welcome any further discussion to ensure all your questions are thoroughly answered. Thank you again for your positive attitude.

---

### Official Review · Reviewer_a726 · 2025-10-31

**Soundness:** 3
**Presentation:** 3
**Contribution:** 3
**Rating:** 6
**Confidence:** 3

**Summary:**

SpeakerVid-5M îs a large-scale dyadic talking humans dataset. The dataset contains automatically extracted annotations of 2D kpts, audio, audio-to-text, and scene descriptions. The dataset is accompanied by an extensive video benchmark.

**Strengths:**

Dyadic human videos are an important mode of human-centric video generation. Having a large dataset with paired ASR, Audio and video is highly valuable. The paper is easy to follow and the authors provide an extensive ethics statement. VidChatBench is a reasonable benchmark suite for the proposed dataset.

**Weaknesses:**

It seems that the dataset only contains pairs of videos - initiator -> respond. However, to future-prove this, I wonder if the authors could make available extended back-and-forth sequences of their data as well, i.e. where the initiator and responder engage in a back and forth way.

The authors claim that the video resolution is 1080P - however, their sample videos in the supplementary material are crops which are significantly smaller than 1080P. Could the authors clarify if they will release the full resolution?

I find Figure 1 a bit unclear: what are the multi-modal annotations? I think the figure should make more clear in what the model inputs and outputs are for the generation process.

It seems some of the example videos contain jump cuts at the end (Body Composition/full_body/3.mp4 ) - I wonder how many videos contain those “”incorrect cuts - could the authors comment if they are planning to post-process the dataset to identify those instances or if they believe those to not be a problem?


Suggested additional citations: TalkCuts [1].


[1] A Large-Scale Dataset for Multi-Shot Human Speech Video Generation; NeurIPS DBT 2025

**Questions:**

Are the person ids per video or are they consistent across multiple videos, i.e. same speaker, different day?

---

> ### Author Response · Authors · 2025-11-19
> **Response to Reviewer a726 (Part 1/2)**
>
> We sincerely thank you for your rigorous and insightful review. Your meticulous observations have been invaluable in helping us strengthen this paper, and we are deeply appreciative of your efforts.  We have thoughtfully considered your comments regarding back-and-forth sequences, resolution, Figure 1, the incorrect cuts, and the person ID, and we respond to each point in turn below.
>
> ### **Response 1: Back-and-Forth Sequences**
> Just as your outstanding insight, the ability to process back-and-forth sequences is fundamental for a model to fully comprehend a conversational context and generate appropriate responses. Our dataset was designed with this principle at its core, and we structure our data to support this requirement specifically. In our paper, this is referred to as the multi-turn dialogue branch, and it is an integral component of our work (detailed in Section 4.5).
>
> Specifically, back-and-forth sequences are represented in our dataset in two distinct formats:
> 1. **Sequential Multi-turn Dialogue**: Each data sample in this category consists of multiple audio-visual clips of an initiator and a responder. These clips are temporally and logically coherent, forming a continuous conversation sequence where the participants interact in a responsive, turn-taking manner. An example of this format is provided in the appendix (Four Data Branch of SpeakerVid-5M/multi-turn), which includes a sample multi-turn conversation.
> 2. **Contextual Multi-turn Dialogue**: However, we recognize the practical challenges posed by long audio-visual sequences, which can lead to a prohibitive number of tokens and high computational complexity. To address this, we introduce a lightweight alternative. This format preserves the responsive initiator-responder structure, but the preceding conversational history is provided solely as ASR text. This design allows a single data instance to be rich in conversational context while substantially minimizing the computational complexity.
>
> ### **Response 2: Video Resolution**
> Our claim regarding the 1080px resolution refers to the fact that the original videos in our dataset are 1080px, and the full-resolution videos are publicly available. The video samples we provide in the supplementary materials are the result of cropping the original footage based on the subjects' spatio-temporal bounding boxes, which accounts for their lower resolution.
>
> In fact, our raw video data can be visually categorized into two distinct formats:
> 1. **Single-Subject Videos**: In this format, only one person appears in the frame. For these instances, the bounding box serves purely as an annotation, and the complete 1080px video can be used directly.
> 2. **Duet-Subject Videos**: This format features both speakers in a dialogue appearing simultaneously within the same frame. To isolate each individual for training, we partition the frame according to their respective bounding boxes. However, frame partitions are also provided in annotation formats, and the full resolution remains 1080px. The user can determine whether to use the full resolution frame or cut the frame with our annotation for their own purpose.
> This process may result in clips with a resolution lower than 1080px for each speaker. This second data type constitutes approximately 20% of the total dataset. For applications with strict high-resolution requirements, users can easily filter to use only the first type, as this distinction is clearly marked in our annotations. However, despite the relatively lower resolution due to frame partitions, duet-subject videos are critically important. The simultaneous presence of both the initiator and the responder is essential for modeling the listener's state, which is a key aspect for creating truly interactive agents (as discussed in Section 4.4). We believe a truly capable digital human must not only be a proficient speaker but also an active listener.

---

> ### Author Response · Authors · 2025-11-19
> **Response to Reviewer a726 (Part 2/2)**
>
> ### **Response 3: Correction of Figure 1**
> Your observation is exceptionally astute, and we are very grateful for your meticulous review and your efforts in helping us further refine our work. We sincerely appreciate your careful identification of the omission in Figure 1, which has now been corrected.
> To clarify, the "multi-modal annotations" depicted apply to a single clip (either an initiator or a responder). These annotations specifically include the corresponding audio, the person ID, structured text labels (e.g., motion intensity, entity lists, expression and action captions), the ASR transcript, the body pose sequence, hand clarity scores, and a full/half-body flag.
> While all the annotations mentioned above are potentially optional inputs, the primary task setting is as follows: the input consists of the initiator's audio and video, along with the responder's initial reference image and a reference voice timbre. The desired output is the synthesized video and audio for the responder. We sincerely apologize for omitting the reference image and reference audio from the input illustration in Figure 1, and we understand the confusion this caused.
> We have carefully revised Figure 1 in the updated manuscript to more accurately reflect the definition of the primary inputs and outputs.
> ### **Response 4: Incorrect Cuts**
> We thank you again for your astute observation and valuable feedback. According to our statistics, approximately 7% of the clips contain incorrect cuts. These incorrect cuts manifest as brief identity changes or abrupt shot jumps at the beginning or end of a clip, with the root cause being the limitations of the original scene detection algorithm used in our initial processing.
> We have already developed and implemented an enhanced filtering protocol to eliminate these artifacts, and the newly cleaned annotations will be incorporated into the updated dataset. To address this issue, we apply four operators in parallel to refine the segmentation of all clips:
> 1. **Identity Consistency**: We use ArcFace to compute the facial similarity between adjacent frames.
> 2. **Scene Consistency**: We extract DINO features for each frame and compute cosine similarity between adjacent frames.
> 3. **Cut Detection**: We compute frame-difference scores between adjacent frames after converting frames to grayscale.
> 4. **Motion Consistency**: We compute motion deltas between adjacent frames based on high-confidence DW-Pose keypoints.
>
> For each video clip, these four operators generate respective frame-to-frame difference scores. By applying a predefined threshold to each score and combining the decisions of all four operators, we achieve a much stricter segmentation of the original clip (usually, only a few frames at the beginning or end are cropped). This ensures that each resulting segment is free from identity changes and shot jumps. Specifically, operators (1) and (2) enforce strict identity consistency, while operators (3) and (4) are designed to resolve the precise issue of abrupt scene changes you identified in the "Body Composition/full_body/3.mp4" example.
>
> ### **Response 5: Person IDs**
> To clarify, the person ID is consistent for a given speaker within a single, continuous source video and does not extend across different recording sessions. Because our source videos are relatively long (0.5–2 hours), all short clips (3–14 seconds) extracted from the same video for a given individual share the same person ID. This design is essential, as it enables us to link consecutive conversational turns from the same individual and thereby organize our multi-turn dialogue data.
>
> ### **Response 6: Suggested Additional Citations**
> We sincerely thank you for your valuable suggestion and for bringing this highly relevant work to our attention. The paper “TalkCuts: A Large-Scale Dataset for Multi-Shot Human Speech Video Generation” presents a high-quality multi-view talking dataset with detailed annotations and strong baseline algorithms. This work has provided us with novel insights and has been highly informative for our research. We have incorporated a citation and a discussion of this work in the related work section of our revised manuscript.

---

### Meta-Review · Program_Chairs · 2026-01-01

**Summary:**

This paper recieved unanimously positive initial ratings. with one strong accept (8) and two borderline accept (6). Reviewers appreciate the valuable large-scale dataset (a726, Ey3j, 8FnX), with thoughtful quality control (8FnX, Ey3j), for a new yet important task (Ey3j) --- audio-visual dyadic interactive digital humans.

Nevertheless, a few key concerns are raised by reviewers:
1. **Dataset scope / generality (Ey3j)**: whether the data distribution (mostly formal/semi-formal scenarios) introduces domain bias that could limit generalization to more “in-the-wild” interaction styles.
2. **Baseline clarity + reproducibility (Ey3j)**: the baseline model description was initially too high-level (Diffusion-MLP details, AR+diffusion loop, definition of “chunk”, token types), raising reproducibility concerns.
3. **Data quality and release details (a726)**: questions about (i) whether multi-turn back-and-forth exists, (ii) whether full 1080p is released vs crops, (iii) prevalence of bad cuts, (iv) clarity of inputs/outputs in Fig.1, and (v) person-ID consistency.
4. **Ethics / legal / bias transparency (8FnX)**: a reviewer flagged concerns about consent/copyright/platform terms and missing demographic breakdowns; additionally, there was concern that this review may contain factual misunderstandings, which complicates how much weight to place on it.
5. **Extending existing pipelines rather than rethinking** (8FnX)

**This paper is conditionally accepted provided the authors do the following for the camera-ready**:
[Ethics] Authors need to detail how they respect copyright and terms of service, and acknowledge any legal risks associated with releasing or using this data. In addition, authors must detail steps taken to remove PII and/or whether people featured in the dataset have consented. They must also discuss any biases in the dataset and its impact on trained models.

**Reviewer Concerns:**

Overall, the author addressed the reviewers' concerns point by point quite well:

1 **Dataset scope / generality (Ey3j)**
* The authors acknowledged the potential bias, and provided justified reason for current choice.
* Though this is not "resolved", the current data collection design is understandable given the expanded reasons.
2.  **Baseline clarity + reproducibility (Ey3j)**
* The authors expanded technical details in terms of e.g., definition of a "chunk", and clarified the AR loop + diffusion refinement and what is fed back. They further implement two offline cascaded baselines for comparison.
3. **Data quality and release details (a726)**
* The authors clarifies that multi-turn exists in two format in current dataset.
* 1080p resolution lies in the raw video source.
* The authors further provided detail statistics for incorrect cuts, and clarify the use of personal IDs.
4. **Ethics / legal / bias transparency (8FnX)**
* The authors provided a quantitative breakdown across gender/age/race/region/language and promised documentation & responsible-use restrictions. They also clarified the release as "annotation + links" (no raw A/V) and argued compliance alignment with YouTube ToS + takedown policy + fair use/GDPR positioning. For safer point, I would like to leave further investigation to ethical reviewers.
5. **Extending existing pipelines rather than rethinking** (8FnX)
* Authors clarified the dataset’s dyadic interactive formulation + four-branch structure + broader viewpoint/body coverage.

**Reviewer Scores:**

I would expect reviewer a726 to slightly increase the rating as their concerns are more clarification-wise, and the authors answers quite clearly. Reviewer Ey3j would likely maintain their score as stated in the post-rebuttal. Reviewer 8FnX holds low confidence. In a fully engaged discussion, if the reviewer accepted the clarifications, I think the most likely outcome is no reliable change (stay ~6) and/or reduced weight in the final decision-making.

Overall, I appreciate all reviewers' effort, and recommend to accept this manuscript.

---

### Decision · Program_Chairs · 2026-01-26

**Decision:**

Accept (Poster)

**Comment:**

Conditions for acceptance have been satisfied.